# Reducing respiratory syncytial virus (RSV) hospitalization in a lower-income country by vaccinating mothers-to-be and their households

Samuel PC Brand[1,2]*, Patrick Munywoki[3], David Walumbe[3], Matthew J Keeling[1,2,4], David James Nokes[1,2,3]

[1]Zeeman Institute of Systems Biology and Infectious Disease Research (SBIDER), University of Warwick, Warwick, United Kingdom; [2]School of Life Sciences, University of Warwick, Coventry, United Kingdom; [3]Epidemiology and Demography Department, KEMRI-Wellcome Trust Research Programme, Kilifi, Kenya; [4]Mathematics Institute, University of Warwick, Coventry, United Kingdom

**Abstract** Respiratory syncytial virus is the leading cause of lower respiratory tract infection among infants. RSV is a priority for vaccine development. In this study, we investigate the potential effectiveness of a two-vaccine strategy aimed at mothers-to-be, thereby boosting maternally acquired antibodies of infants, and their household cohabitants, further cocooning infants against infection. We use a dynamic RSV transmission model which captures transmission both within households and communities, adapted to the changing demographics and RSV seasonality of a low-income country. Model parameters were inferred from past RSV hospitalisations, and forecasts made over a 10-year horizon. We find that a 50% reduction in RSV hospitalisations is possible if the maternal vaccine effectiveness can achieve 75 days of additional protection for newborns combined with a 75% coverage of their birth household co-inhabitants (~7.5% population coverage).

*For correspondence:
S.Brand@warwick.ac.uk

**Competing interests:** The authors declare that no competing interests exist.

## Introduction

Respiratory syncytial virus (RSV) is the most common viral cause of acute lower respiratory infection (*Nair et al., 2010*). A large majority of children contract RSV by the age of two (*Glezen et al., 1986*; *Ohuma et al., 2012*), but the chance of developing severe disease from a RSV infection is much greater amongst young infants (6 months) (*Hall et al., 2009*) and decreases rapidly with the age of the infected child. Vaccine development aimed at protecting young children against RSV disease has become a global health priority (*World Health Organization, 2017*). As of December 2018, there are over 40 RSV vaccines in development (*PATH, 2018*). In particular, two vaccination approaches have been identified as potentially effective: a single dose vaccine aimed at mothers-to-be leading to antibody transfer across the placenta thereby boosting maternally acquired immunity among newborns, and paediatric vaccination aimed directly at infants (*Modjarrad et al., 2016*; *World Health Organization, 2017*). Moreover, it is possible that a prophylactic extended half-life monoclonal antibody could act as a vaccine surrogate whilst replicating the desired effect of a maternal vaccine (*Zhu et al., 2017*; *Domachowske et al., 2018*). A serious complication in RSV vaccine development has historically been the risk of causing enhanced disease amongst the immunologically naive (*Chin et al., 1969*), therefore it might be more prudent to target a paediatric vaccine at older children with better developed immune systems rather than young infants most at risk of RSV disease (*Anderson et al., 2013*). Epidemiological data suggests older individuals (elder siblings, parents) are

potential sources of infection for the infant of the household (*Graham, 2014*), for whom temporary boosted immunity might best be achieved using a sub-unit vaccine (*Anderson et al., 2013*).

The desired effect of vaccinating older children is two-fold: the vaccine both decreases the risk of morbidity in the vaccinated child and reduces the risk of transmission from the older child to any young infant the vaccinated child contacts (*Anderson et al., 2013*). Molecular analysis of nasopharyngeal samples collected from a semi-rural community in Kenya has identified that the majority of RSV infections among young infants originated from within their household rather than the wider community, with older siblings being the usual household index case (*Munywoki et al., 2014*), echoing a previous household study of RSV transmission (*Hall et al., 1976*), although it should also be noted that the young infant was herself the index case on a significant number of occasions. This finding emphasises that reducing transmission to young infants within the household could be an effective way of reducing RSV disease in low- and middle-income countries (LMICs). However, the significant number of young infant index cases within households suggest that 'cocooning' young infants from transmission by vaccinating others in their household may not be sufficient by itself. Ideally, cocoon protection should be achieved in conjunction with directly protecting the young infants using a maternal vaccine.

At this time, the only reported phase III trial on RSV vaccine effectiveness is for the maternally targeted ResVax, which failed to meet its primary objective but nonetheless showed partial effectiveness at reducing hospitalisations due to RSV (*NovaVax, 2019*). The possibility that a vaccine for only one target population might be only partially effective, and the importance of RSV transmission within the household, motivates our modelling approach. In this paper, we assess the efficacy of a mixed vaccination strategy in a LMIC setting, Kilifi county Kenya. In our scenarios, there was at least one maternal vaccine and one paediatric vaccine available as per WHO priority (*World Health Organization, 2017*). In Kenya, there are very high rates of prenatal contact between pregnant women and health professionals (97.5% in Kilifi county; *KNBS, 2015*). This suggested targeting pregnant women as part of their prenatal contact, and then offering the paediatric vaccine to all over one year olds, including adults, cohabiting with the pregnant mother. The essential idea was to leverage prenatal contact to achieve a very high coverage of a maternal antibody boosting (MAB) vaccine, and also to target her household cohabitants with an immune response provoking (IRP) vaccine. The IRP vaccine elicits an immune response and, therefore, a temporary reduction in susceptibility to RSV for the vaccinated individual. We follow (*Yamin et al., 2016*) in assuming that the elicited period of immunity to RSV from receiving the IRP vaccine would be similar to that of a natural infection.

Predictions of vaccine effect are derived from a dynamic transmission model designed to capture the demographic structure of the population, the seasonality of RSV transmission and how rapidly, and to whom, RSV is transmitted in both households and the wider community. Unknown model parameters were inferred using data from the large-scale long-running Kilifi Health and Demographic Surveillance System (KHDSS; *Scott et al., 2012*), and hospitalisation admissions at Kilifi county hospital (KCH) confirmed as due to RSV since 2002. It should be noted that targeting vaccination in this way is not an approach that one would expect to greatly reduce RSV infections under the assumptions of simple compartmental models of RSV transmission because the rate of vaccination deployment would be too low (see *Box 1*). However, we shall see that these vaccines are efficiently targeted at creating protection for the young infants most at risk of hospitalisation if they caught RSV.

The modelling approach used in this paper differs from the majority of RSV modelling approaches extant in the literature, which largely focus on deterministic age structured transmission models (*Pitzer et al., 2015*; *Kinyanjui et al., 2015*; *Yamin et al., 2016*; *Hogan et al., 2016*). In contrast, we explicitly model the social clustering of individuals into households. The advantage of explicit inclusion of household structure in the model is that the social contacts within the household are persistent over multiple RSV seasons, whereas age-structured models implicitly assume random mixing; that is all people of a given age group are equally likely to be contacted by any individual at any instant and therefore the chance of repeated contact become zero as the population size becomes large. In the specific case of modelling highly seasonal RSV transmission, it is likely that capturing the network-like transmission structure of the population is important for representing the relevant epidemiology. Most people have caught RSV by the age of two, and will have multiple repeated episodes during their lifetime. The time between recovery from an episode and reversion back to at least partial susceptibility is estimated to be 6 months (*Ohuma et al., 2012*). In Kilifi county, there

## Box 1. Vaccination predictions from a simple unstructured RSV epidemic model.

The essential idea in this paper is to use prenatal contact between mothers-to-be and health professionals to deploy two separate vaccines: first, a vaccine targeting the mothers-to-be which boosts the duration of protection her newborn will have against RSV (MAB vaccine), and second, a vaccine aimed at the mothers-to-be's household cohabitants giving each a period of RSV immunity, equivalent to that of a natural infection (IRP vaccine). As a baseline for understanding RSV transmission we can use a simple mechanistic model which captures the essential biology of RSV infection; newborns are born with a period of immunity to RSV infection which is lost during their first year of life, after contracting RSV the individual is infectious for a period before gaining temporary waning immunity to RSV re-infection. Assuming homogeneous transmission the dynamics of the simple RSV transmission model can be described using four dynamic variables describing the numbers of currently maternally protected individuals (M), susceptibles (S), infecteds (I) and immune/recovereds (R). The evolution of the epidemic, after vaccination, can be given as a standard ODE:

$$\dot{M} = B - \alpha_{vac}M - \mu M, \; \dot{S} = \alpha_{vac}M - \frac{\beta}{N}SI + \nu R - \mu S - B\langle H\rangle V_{cov}\frac{S}{S+I+R},$$
$$\dot{I} = \frac{\beta}{N}SI - \gamma I - \mu I, \; \dot{R} = \gamma I + B\langle H\rangle V_{cov}\frac{S}{N} - \mu R - \nu R.$$

where each term above describes the rate of events that change the epidemic state: Births ($B$), loss of maternally derived protection after MAB vaccination, ($\alpha_{vac}$), mortality ($\mu$), RSV force of infection ($\beta I/N$), recovery ($\gamma$), reversion to susceptibility ($\nu$), as standard in the literature (**Anderson and May, 1991**; **Keeling and Rohani, 2008**). The rate at which IRP vaccines successfully vaccinate susceptibles is $B\langle H\rangle V_{cov}S/(S+I+R)$; that is the mean size of a pregnant woman's household ($\langle H\rangle$) times the effective coverage of the vaccine ($0 \leq V_{cov} \leq 1$) time the likelihood of selecting a susceptible and not wasting the vaccine assuming that we are only targeting those who have definitely lost their maternal protection to RSV ($S/(S+I+R)$). For simplicity, we can treat the duration of maternal protection as very short compared to the typical person's lifetime (i.e. $\alpha_{vac} \gg \mu$). The equilibrium of the simple RSV model is analytically tractable (see appendix 2):

$$\text{Relative reduction in transmission due to vaccination} = \frac{\mu\langle H\rangle V_{cov}}{(\nu+\mu)(R_0-1)}$$

$$\text{Reduction in transmission per IRP vaccine} = \frac{\gamma+\mu}{R_0(\gamma+\mu+\nu)}$$

where $R_0 = \beta/(\gamma+\mu)$ is the reproductive ratio of RSV, and we are assuming that the birth rate is at replacement $B = \mu N$. The simple RSV model makes some general predictions about the efficacy of IRP vaccination:

Therefore, a naive simple model of RSV transmission is pessimistic about the joint vaccination strategy. However, in this study, we also account for more detailed social structure, differential susceptibility, infectiousness, and risk of disease dependent on the age of the individual and seasonality in transmission. We will see that targeting vaccines socially close to young infants is much more effective than the simple model predicts.

- The MAB vaccine does not significantly effect transmission in the general population.
- The efficiency of the IRP vaccine (avoided infections per effective dose) should not change with coverage.
- Using parameters typical of the study population at Kilifi (see appendix 2), the reduction in RSV transmission due to IRP vaccination can be modest because the deployment rate is too low; for $R_0 = 2$ the maximum achievable reduction in transmission is < 4% compared to no vaccination.

are sharp annual peaks of RSV hospitalisation at each seasonal RSV epidemic, and so one should expect the population to consist of large numbers of entirely susceptible individuals, who have never caught RSV before and are primarily in their first 2 years of life, and partially susceptible individuals, who have caught RSV at least once before, due to the inter-epidemic period being longer than the typical time over which loss of immunity to RSV occurs. These general considerations suggest that (i) RSV seasonal epidemics will be akin to repeated invasions of a nearly susceptible population, that is closer to an epidemic scenario than an endemic scenario, and (ii) RSV transmission is much closer to a SIS rather than a SIR paradigm. Social network effects in epidemiological forecasting are most important during an epidemic invasive growth phase and are typically more important for SIS-type dynamics with persistent contacts (*Miller, 2009*; *Sun et al., 2015*). Both these features appear to be important for seasonal RSV transmission in Kilifi and therefore provide strong motivation for the network-type epidemic model we have used.

Two possible explanations for the comparative lack of using household structure in RSV modelling are: first, accounting for the interplay of demography and household structure remains a significant modelling challenge (*Glass et al., 2011*; *Geard et al., 2015*), and second, the dynamics of age structured transmission models can be predicted using a comparatively small set of deterministic rate equations (*Keeling and Rohani, 2008*). Moreover, whenever natural immunity is long-lasting and/or high levels of effective vaccination coverage exist for the population, household structure is less important and can be captured using simple approximations, for example, the mother-child contact approximation (*Atkins et al., 2016*). As a possible alternative modelling framework stochastic individual-based models (IBMs) for epidemics benefit from additional realism and flexibility compared to deterministic models, and there does exist at least one modelling study considering the effect of social structure on RSV transmission using a non-seasonal approximation within a stochastic individual-based model (IBM) (*Poletti et al., 2015*). However, rigorous inference of model parameters for stochastic IBMs of epidemics is highly challenging because, along with other difficulties, the random infection times of each case will not typically be known (*O'Neill and Roberts, 1999*). The model used in this paper required a rate equation for each possible household configuration (*House and Keeling, 2008*). Specifically for RSV modelling it has been noted that this could lead to thousands of rate equations that must be simulated simultaneously (*Kinyanjui, 2014*), effectively rendering the model impractical for regression against data due to slow integration time. Nonetheless, this work demonstrates that by making appropriate simplifications, and using numerical solvers adapted to large systems (in this case ~2000 variables), it was possible to both include realistic household structure and rigorously infer model parameters for a model of RSV transmission in a LMIC setting.

## Results

The RSV transmission model parameters were either drawn from the RSV literature or inferred from age-stratified weekly hospitalisations at Kilifi county hospital (KCH) between 2002 and 2016. The underlying biology of the transmission model was similar to a simple compartmental model of RSV infection and waning immunity (see *Box 1*) with two main differences: (i) the age of the individuals affected their susceptibility to RSV, infectiousness after contracting RSV, duration of RSV infectiousness, and likelihood of developing severe disease and being hospitalised after contracting RSV, partly because of age-specific effects, and partly because we assumed that every person had caught RSV at least once after their first year of life, and (ii) infectious contacts were distributed at two levels of social mixing differentiating between persistent contacts between household co-occupants and randomly assigned contacts within the community of Kilifi county based on the ages of the infected and infectee (*Figure 1* and Materials and methods). The joint age and household distribution of the population accessing KCH was chosen to match the ongoing findings of the Kilifi Health and Demographic surveillance system (KHDSS; *Scott et al., 2012*). The seasonality of RSV hospitalisations at KCH has historically been erratic with peak months for RSV hospitalisation varying as widely as November to April (appendix 1). Moreover, over the 15-year period we are studying in this paper, there was demographic change in the underlying population both in age profile and household size distribution. We addressed these modelling challenges: first, by rejecting the typical epidemiological modelling assumption that population demographic structure is at equilibrium in favour of directly modelling demographic change, and second, by treating the shifting seasonality of RSV transmission in Kilifi as being driven by an underlying latent random process to be jointly

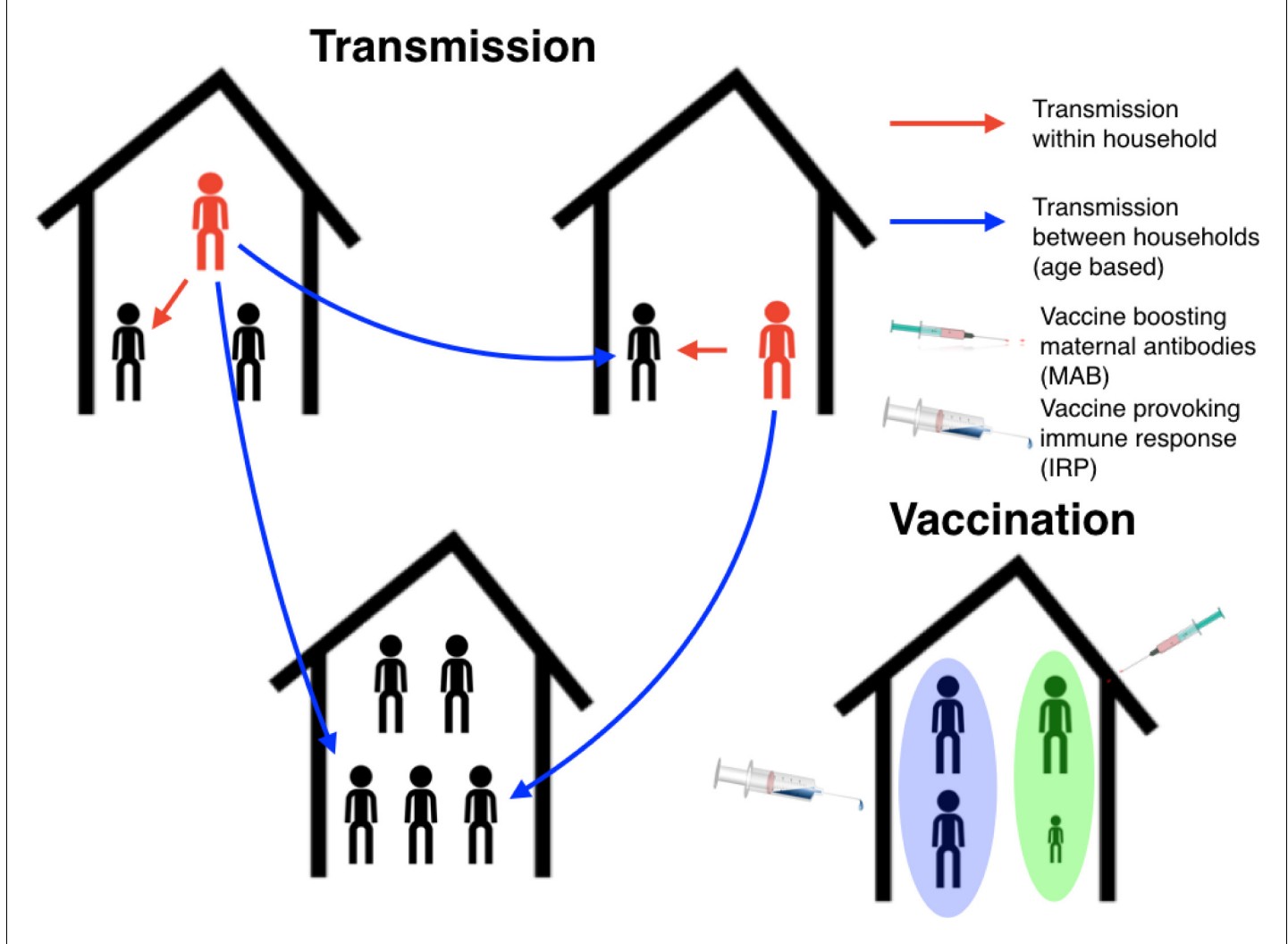

**Figure 1.** Schematic plot for the RSV transmission model and vaccination programme. Infectious individuals (red character figures) transmit to other individuals inhabiting the same house, and to other individuals in other households based on the ages of the both the infector and infectee. Red and blue arrows represent possible realised infections over a short period of time. Bottom right household demonstrates the vaccination strategy; the mother has received a maternal antibody boosting (MAB) vaccine which increased transfer of protective antibodies to newborns (green background shading), meanwhile other household members have received an immune response provoking (IRP) vaccine (blue background shading).

inferred with model parameters. The goal was to account for factors influencing the rate of hospitalisations that changed over the 15 years of study so as to get an unbiased estimate of parameters we assumed were static over the period, such as the person-to-person rate of transmission within a household. We were able to broadly capture the year-to-year variation in hospitalisation, and age profile of the hospitalised, with only six free parameters (*Figure 2*, Materials and methods, and appendix 1). The 2005/2006 RSV year (see appendix 1 for RSV year definition) was anomalous in that there were three peaks in RSV hospitalisation separated by at least a month: two smaller peaks on 11th Dec 2005 and 24th Mar 2006 around a larger peak on 24th Feb 2006. The model was unable to explain this unusual year, other years having solitary peaks. Outside of the 2005/2006 RSV year there were 2174 hospitalisations during the period of study compared to a model prediction of 2147 hospitalisations ([2057, 2238] 95% prediction interval ). We were unable to jointly identify the rate of school children contacting other school children with the rate of homogeneous contact among all over one year olds, therefore we considered a range of within school contact rates, and for each value inferred the other six free model parameters and assessed the efficacy of vaccination for a range of MAB vaccine effectiveness values and IRP vaccine coverage values. Each scenario gave

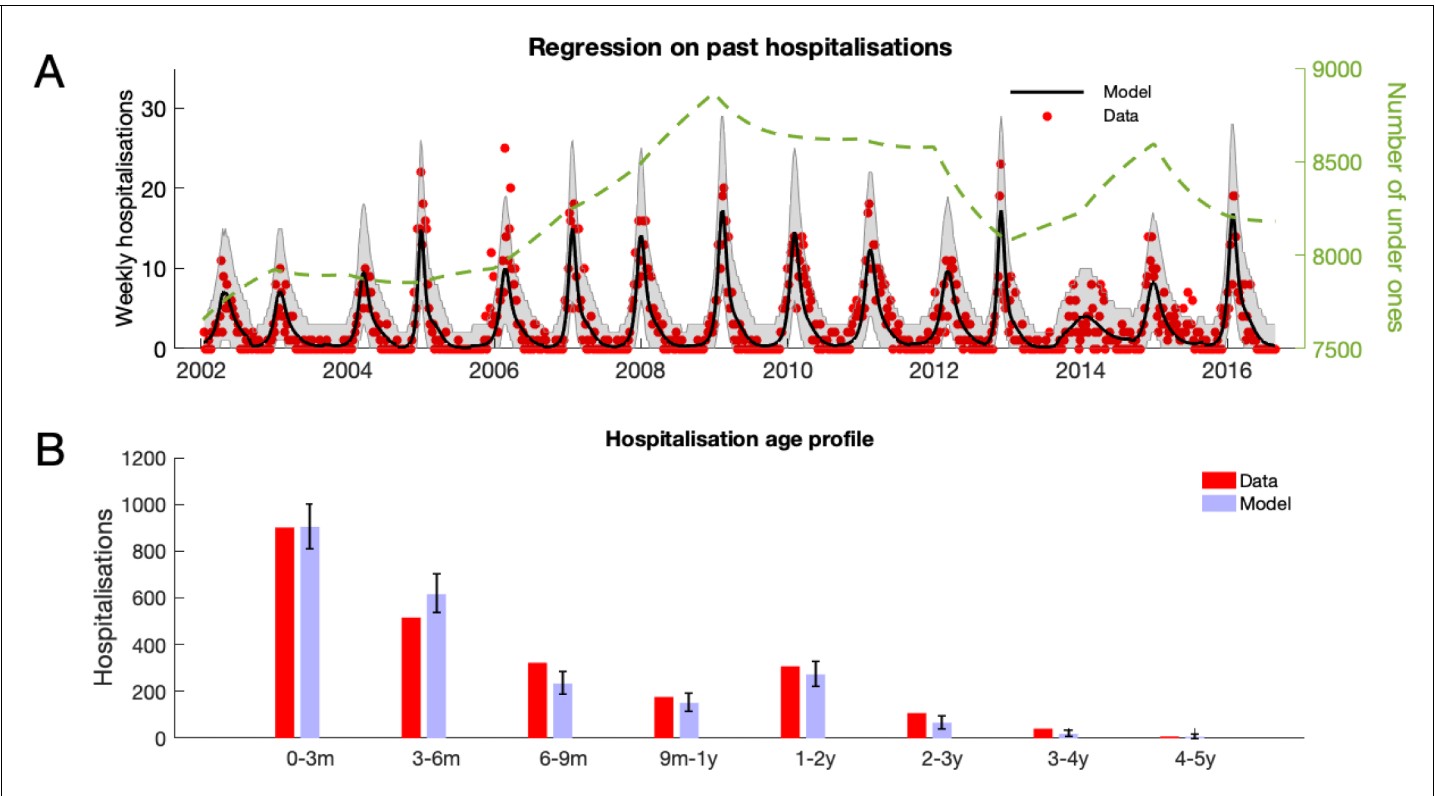

**Figure 2.** RSV hospitalisation at KCH: dynamics and age profile of hospitalised patients. (**A**) Weekly RSV hospitalisations before implementation of vaccinations. Black curve gives mean prediction of RSV household transmission model after regression against weekly incidence data (red dots). Grey shaded area indicates the 99% prediction interval for the model. Also shown is the number of under ones in the population (dashed line). (**B**) Age profile of hospitalisations at KCH before vaccination. Error bars give 99% prediction intervals for model.

The online version of this article includes the following source data for figure 2:

**Source data 1.** Hospitalisation data, and model predictions, are given as MATLAB data files along with script for plotting figure.

similar results for the efficacy of household targeted vaccination (see appendix 3), therefore we have only presented results in the main *Results* section for the scenario with the highest rate of within school mixing. At KCH all RSV hospitalisations occurred in the under five year olds with 84% of hospitalisations occurring in the under 1 year olds (*Figure 2B*). This finding is consistent with the much higher rates of hospitalisation per RSV infection for younger infants (*Kinyanjui et al., 2015*). However, the hospitalisation time series has to also be understood in the context of dynamic RSV transmission and demographic change in the study population. A general trend of increasing hospitalisations between 2002–2009 is at least partially explained by a 16% increase in under ones in the population over that period. The rest of year-to-year variation in hospitalisation was explained by seasonal epidemic dynamics, themselves driven by shifting seasonality (*Figure 2A*; 1).

We found that, pre-vaccination, school age children suffered on average the highest force of infection, that is the per-capita rate of infectious contacts, from outside of the household followed by under 1 year olds (*Figure 3A*). This finding was dependent on assuming that we had a high degree of homophily in the social contacts of school-age children (the high within school transmission scenario mentioned above). Other scenarios were considered with lower levels of in-group preference for school-age children to contact other school-age children; in the alternate scenarios, the parameter imputation process found slightly higher rates of contacts within the household and homogeneously outside of the household but lead to very similar results (appendix 3 ). The infectious contacts outside the household were distributed predominantly to individuals within households of size 2–5 (*Figure 3*). This reflected the household distribution of the population; school

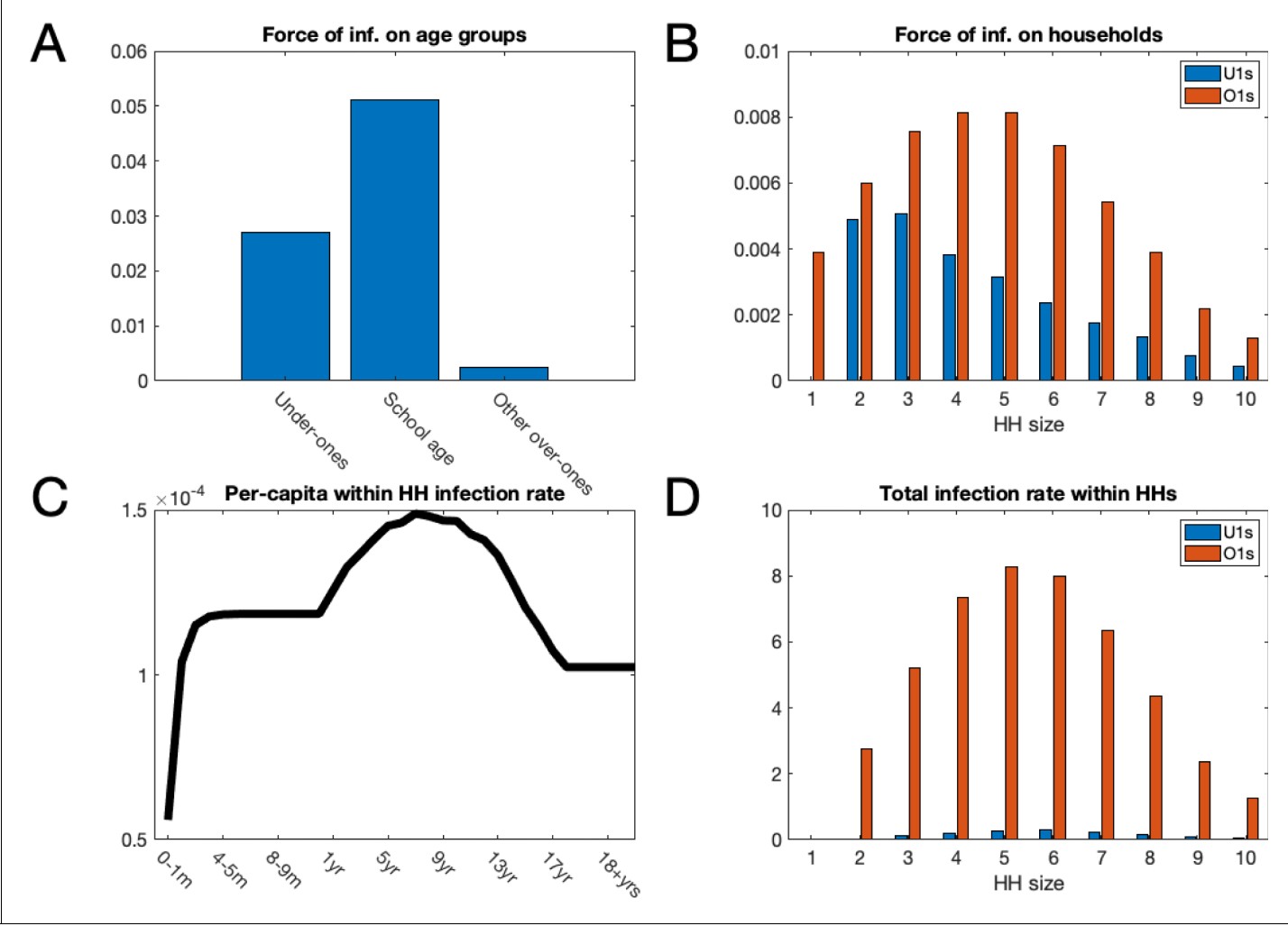

**Figure 3.** Mean force of infection (2002–2016) between households and mean infection rates within households. (A) The mean force of infection (infectious contacts received per person per day) of RSV due to transmission from without the household on three age groups: under-ones, school age children and everyone else, including adults. (B) Mean force of infection due to transmission without the household on individuals inhabiting each household size. (C) The mean per-capita daily rate at which different age groups become infected with RSV from within their household. (D) The mean total daily rate of RSV infection within households of different sizes.

The online version of this article includes the following source data for figure 3:

**Source data 1.** The model predictions are given as MATLAB data files, along with the script for plotting figure.

children and under ones who were most at risk of making social contact with those infected with RSV outside the household tended to live in households of this size (*Figure 3B*).

Force of infection is a less natural concept for measuring within household infection due to small numbers of individuals per household, and intense frequent contacts. Instead, we measured the true rate of RSV transmission between individuals cohabiting a household. The highest per-capita rates of infection within households were for 7 year olds (*Figure 3C*); this reflected the typical age of individuals within the households most at risk of RSV introduction and with severest transmission rates after introduction. The infection rate among under ones increased rapidly until it plateaued at ~6 months old. The rapid increase in per-capita infection rate was due to waning of maternally acquired immunity to RSV, which we inferred as lasting on average 21.6 days ([17.2, 26.1] 95% CI; see Table 3 for all inferred parameters). The total infection rate within households was greatest in size 5 and 6 households (*Figure 3D*). This differed from the household size where each person was at most risk of contracting RSV outside the household. Two factors shifted the burden of RSV infection to larger

households: first, there are more people in larger households therefore risk of RSV introduction can be higher even if the per-person rate is lower, and second, the intensity of transmission within households is higher for larger households.

We evaluated a series of scenarios where a combination of a maternal antibody boosting (MAB) and an immune response provoking (IRP), vaccine were targeted at, respectively, mothers-to-be in their third trimester, and their household cohabitants upon the birth of the newborn. Between scenarios we varied (i) the effectiveness of the MAB vaccine, (ii) the coverage of the MAB vaccine, and (iii) the household coverage of the IRP vaccine, see *Table 1* for a list of all vaccination scenarios modelled in this paper. The protective effect of the vaccines on individuals was the same as for the unstructured population model presented in *Box 1*: the MAB vaccine increased the period over which a newborn was protected from RSV by maternally acquired antibodies, and the IRP vaccine, given to all household cohabitants of some participating mothers-to-be, initiated an immune response in the vaccinated which gave a period of protection from acquiring RSV similar to that following a natural infection. The high prenatal contact levels in Kilifi county suggested that vaccination coverage of mothers-to-be had the potential to be very high, especially if maternal immunisation to boost newborn immunity became an established method for a range of vaccines including influenza and Group B Streptococcus. However, an available MAB vaccine might only be effective if delivered in the third trimester of pregnancy and, whilst having at least one prenatal contact is very common for pregnant women in Kilifi county, it is not clear that prenatal contact always occurs at the relevant stage of pregnancy. Therefore, we consider both an optimistic scenario (100% MAB coverage), and a more conservative uptake (50% MAB coverage). The number of days of additional maternally derived protection donated to the newborns by MAB vaccinated mothers was uncertain, we considered a range of MAB protection 0–90 days. We assumed that if the pregnant mother's household cohabitants agreed to receive an immune response provoking vaccine then all were vaccinated at the birth of the newborn to maximise the overlap between the protection period of the cohabitants and the first months of life of the newborn. As is common in vaccine strategy analysis, we combine coverage and effectiveness into one effective coverage (coverage times effectiveness c.f. *Keeling and Rohani, 2008*), although in this case effective coverage could be considered both within and between households.

We assumed that the maximum coverage of the vaccine would be reached within a year, and considered 10 years of RSV transmission after this implementation. When inferring model parameters we took care to account for the known changes in demography over the study period, both in the age and the household occupancy distributions of the population. However, for the 10-year forecasting in this paper, we assumed that the total birth rate was constant (8601 per year), and that the population age and household occupancy distributions remained static. The model inference stage included inferring the statistics of yearly variation in RSV seasonality. The decrease in rates of RSV hospitalisation and infection due to vaccination over ten years presented are median improvements over 500 independent realisations of random future seasonal patterns compared to a baseline of no intervention. If the MAB vaccine was unavailable or ineffective (0 days MAB protection), we found that it was still possible to reduce RSV hospitalisations by up to 25% using only the IRP vaccine on the household members of young infants at time of birth (*Figure 4A and B*). If 100% maternal vaccination could be achieved then the MAB vaccine was more successful as a sole vaccine option compared to IRP vaccination; in the sense that 90 days of additional protection from RSV delivered a 45% reduction in hospitalisation even with no IRP vaccine coverage. Nonetheless, even with an

**Table 1.** Modelled vaccination scenarios.

Each combination of MAB vaccine effectiveness and coverage, with IRP vaccine coverage below was one scenario. The baseline scenario being no effective MAB vaccine and 0% coverage of IRP vaccine.

| Description | Range |
|---|---|
| Additional period of protection from RSV at birth due to maternal antibody boosting (MAB) vaccine ($P$). | 0 (no vaccine), 15, 30, 45, 60, 75, 90 days |
| Coverage of mothers with MAB vaccine | 50%, 100% |
| Coverage of households with newborns with immune response provoking (IRP) vaccination ($V_{cov}$) | 0%, 25%, 50%, 75%, 100% |

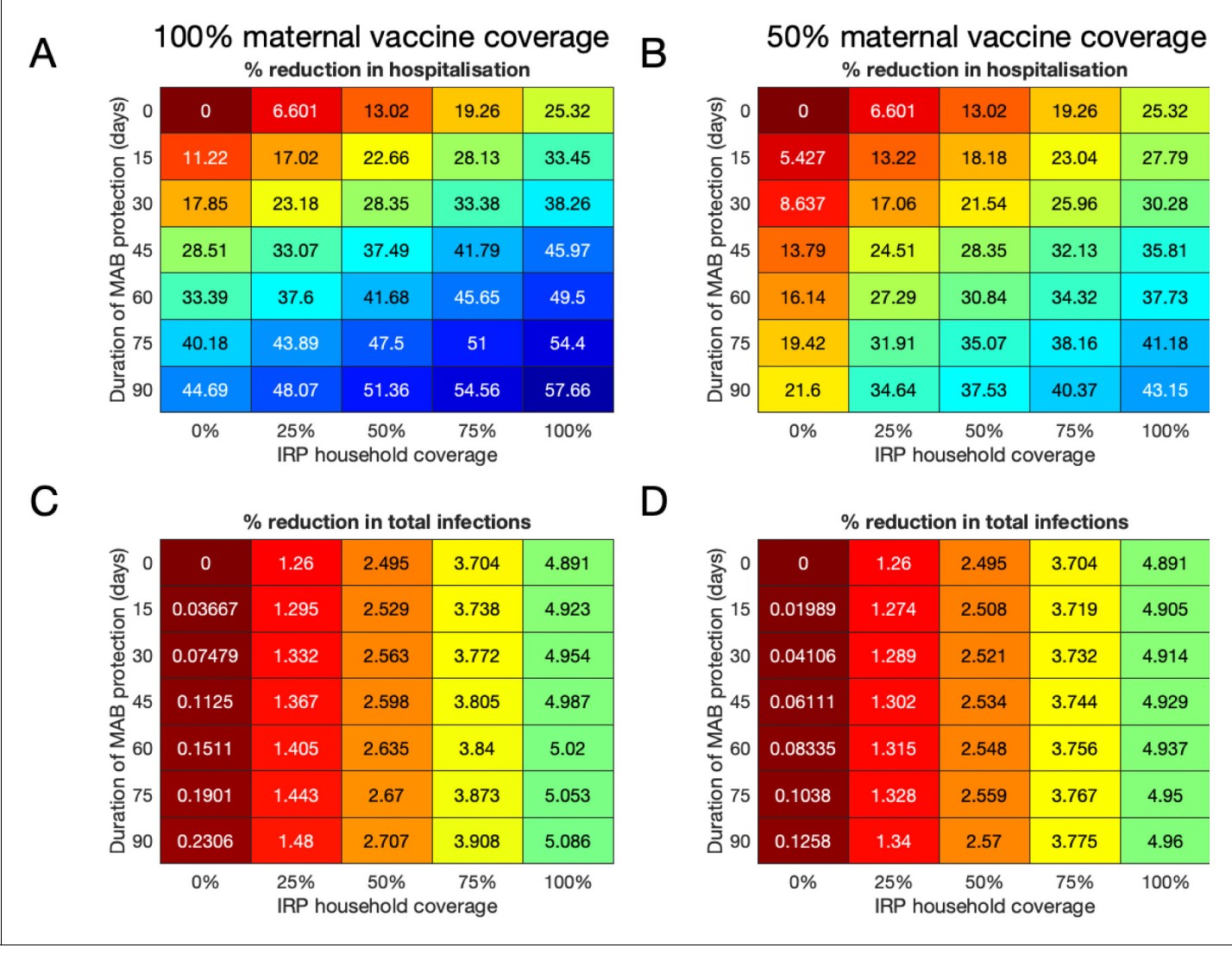

**Figure 4.** Median forecast effectiveness of RSV vaccination for different mixed strategies over a 10-year period for 100% maternal vaccine effective coverage (A and C) and 50% maternal vaccine effective coverage (B and D). (A and B) Median percentage reduction in hospitalisations at KCH. (C and D) Percentage reduction in total RSV infections in the population.

The online version of this article includes the following source data for figure 4:

**Source data 1.** Reductions in hospitalisations and infections for each of the 500 forecasting simulations are given as MATLAB data files, along with script for plotting figure.

effective MAB vaccine there was added benefit to also using a IRP vaccine; a greater than 50% reduction in hospitalisations was achieved with a MAB vaccine that gave 75 additional days of RSV protection and a 75% coverage of the pregnant womens' households (*Figure 4A*; a colorblind-friendly version of this plot can be found as appendix 4 Fig D). If only 50% maternal vaccination coverage could be achieved then unsurprisingly also using the IRP vaccine became relatively more important. The mixed vaccination strategy that achieved better than 50% hospitalisation reduction with 100% maternal coverage achieved 38% reduction in hospitalisations with 50% maternal coverage (*Figure 4B*); halving the maternal coverage didn't necessarily halve the success of the vaccination programme so long as IRP vaccine was also available. Improving the effectiveness of the MAB vaccine caused a significant improvement in hospitalisations, but had an almost negligible effect on the total infections in the population (*Figure 4C and D*). IRP vaccination was more effective at reducing total RSV infections, but even at 75% coverage of the households of women giving birth the reduction in infections was <4% (*Figure 4C and D*). That IRP vaccination had a modest effect on the

true infection rate, and that MAB vaccination has a negligible effect on the true infection rate, was in line with the prediction of the simple non-seasonal RSV model (*Box 1*). However, the simple model cannot predict that the percentage reduction in hospitalisations would be significantly greater than for total infections because of the direct and indirect protection of those most at risk of disease. For the mixed strategy achieving a 50% reduction in RSV hospitalisations described above (75 days direct MAB protection at 100% MAB coverage with 75% IRP household coverage), the seasonal dynamics of hospitalisations post-vaccination equilibrated rapidly (*Figure 5A*). There was a reduction in median hospitalisations in every age group, but predominantly in 0–3 month years old (who are nearly all protected by the MAB vaccine) and 3–6 month year olds (*Figure 5B*). However, targeting pregnant women and their cohabitants did not prevent sufficient RSV infections as to significantly disrupt RSV transmission within the population at large, which may explain the rapid approach to new RSV hospitalisation dynamics. Nonetheless, those who were protected were overwhelmingly among those at most risk of disease if they had caught RSV.

Each vaccine used decreased the expected number of RSV infections and hospitalisations. As well as measuring the overall effectiveness of RSV vaccination (see above), we also measured the efficiency of vaccination, defined as number of infections or hospitalisations averted per vaccine (of either type). Unsurprisingly, as the duration of protection given by the MAB vaccine increased the efficiency of vaccination also increased; significantly for hospitalisations (*Figure 6A*) and marginally for infections (*Figure 6B*). This was true whether an IRP vaccine was used, or not. If there is no MAB vaccine available then the efficiency of using only IRP vaccination doesn't change with coverage; that is that when increasing IRP household coverage the improvement per vaccine used stayed static, in line with what one might expect from a homogeneous mixing RSV model (see *Box 1*). However, when MAB and IRP vaccines were used in conjunction there was an efficiency penalty due to redundancy in the each vaccine's protective effect. For example, if a MAB vaccine was available that gave 90 days protection the marginal benefit in terms of decreased hospitalisations of having an IRP vaccine was decreased because most at-risk infants were already protected by the MAB vaccine

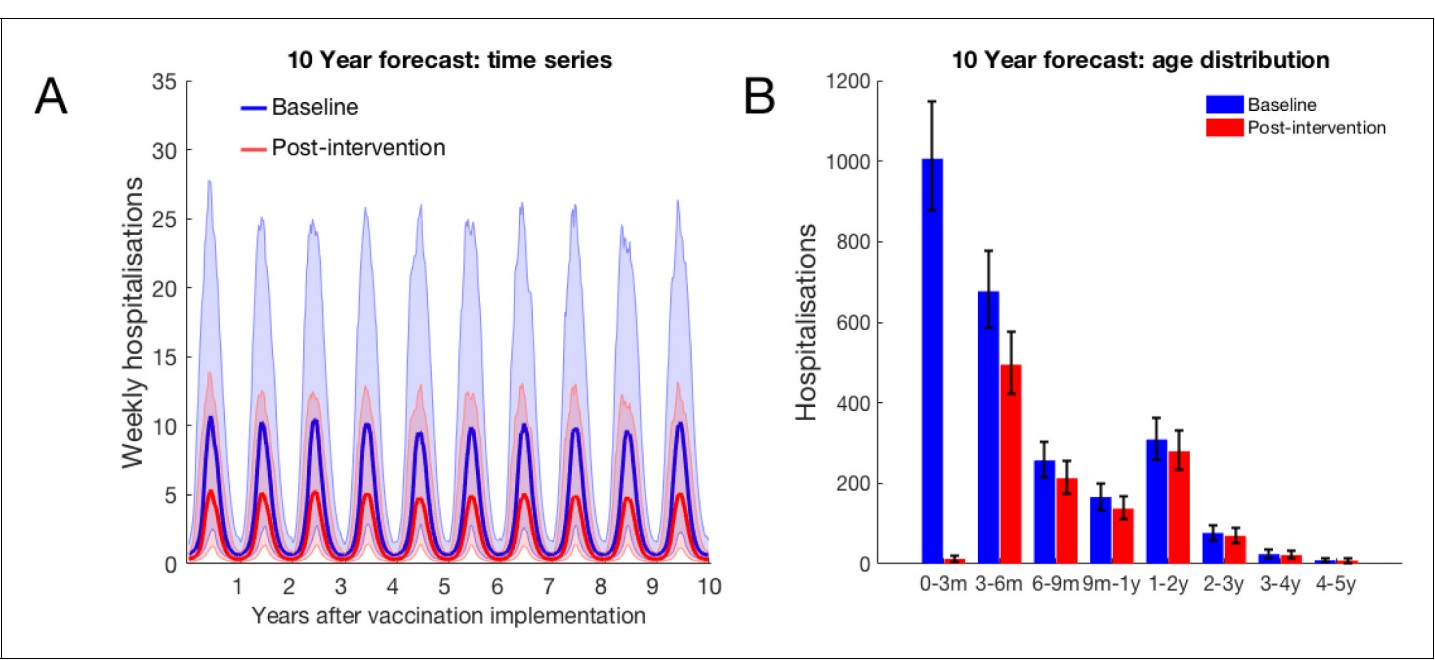

**Figure 5.** 10-year forecast of RSV vaccination effectiveness for a mixed strategy of an MAB vaccine provided 75 days of additional RSV protection for newborns and a 75% IRP vaccine household coverage. (A) Forecast weekly hospitalisations for a baseline of no vaccination (*blue*) and the mixed vaccination strategy (*red*). Shown are median forecast (*curves*) and 95% prediction intervals (*background shading*). (B) Forecast age distribution of total RSV hospitalisations at KCH. Median forecast (*bars*) and 95% prediction intervals (*error bars*).

The online version of this article includes the following source data for figure 5:

**Source data 1.** Hospitalisation predictions for each of 500 forecasting simulations is given as a MATLAB data file, along with a MATLAB function for combining the forecasting and Poisson hospitalisation rate uncertainties into a prediction interval and plotting script.

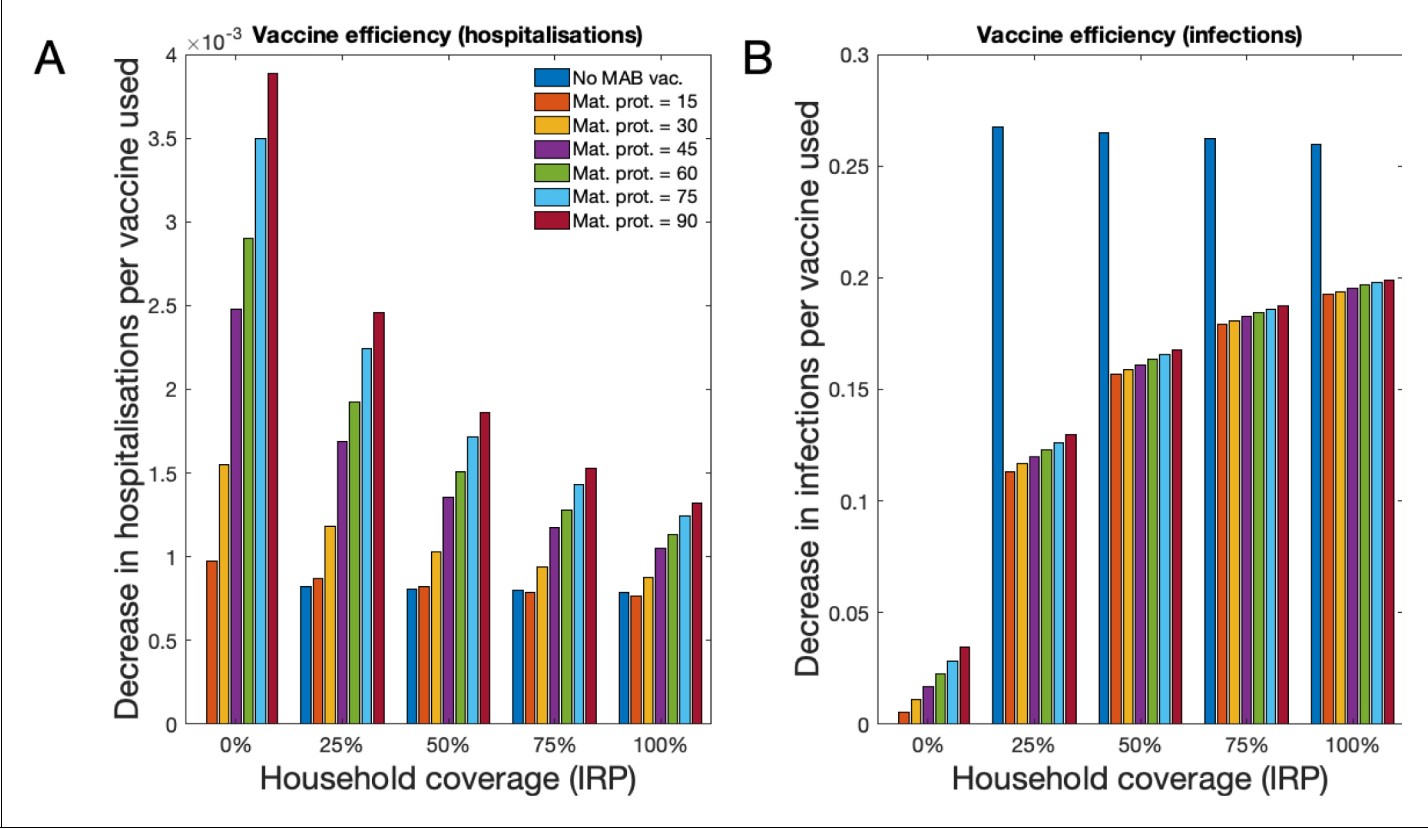

**Figure 6.** Forecast vaccination efficiency against hospitalisations and all infections, defined as number of cases averted per vaccine used (both MAB and IRP). MAB vaccine coverage was 100% unless unavailable, however MAB protection duration varied (different coloured bars) and IRP household coverage was also varied. See *Table 1* for a list of scenario. (A) Median avoided hospitalisations at KCH per vaccine over 500 simulations. (B) Median avoided RSV infections in population per vaccine over 500 simulations.

The online version of this article includes the following source data for figure 6:

**Source data 1.** A MATLAB script for converting 500 forecasting simulation outcomes into efficiency metrics, and plotting them.

(*Figure 6A*). Using two types of vaccine always decreased infections and hospitalisations (see above), but the total reduction was always less than simply adding the reductions of each vaccine in the absence of the other.

## Discussion

Our modelling analysis suggested that a high-coverage vaccination campaign of mothers-to-be with a vaccine inducing elevated levels of transplacenta RSV antibody transfer to her newborn, alongside targeting the newborn's cohabitants with a generic vaccine that provoked a period of immunity to RSV can achieve greater than 50% reduction in hospitalisations due to RSV. This combined vaccination strategy suggested itself due to the high prenatal contact rates between mothers-to-be and health professionals in Kilifi county, Kenya (97.5% **KNBS, 2015**). We found that the combined vaccination strategy was efficient at targeting effort towards directly protecting young infants most at risk of developing RSV disease with boosted antibodies, and filling in any gap in protection with indirect cocoon protection within the household using a vaccine aimed at older cohabitants. Even at maximum effective household coverage for the IRP vaccination only ~10% of the population were vaccinated each year with a modest reduction in the RSV infection rate of ~5%. Nonetheless, at that coverage IRP vaccination alone achieved a 25% reduction in hospitalisations at KCH even without an effective MAB vaccine to provide direct protection to young infants. This demonstrated that although we were vaccinating at a low rate compared to population size, with only a modest reduction in infection rate, those people we did vaccinate were efficient at cocooning young infants from

transmission and therefore risk of severe disease. If an effective MAB vaccine was also available the reduction in hospitalisations was greater, although the additional protection due to cocooning was relatively less since young infants were also protected from contracting RSV at the age when they were at most risk of severe disease.

We constructed the model used in this paper with the purpose of estimating the efficacy of targeting pregnant women and their households for vaccination. In order to make predictions mechanistic models of disease transmission must approximate the social structure of the population being modelled, and hence the contact rates between individuals. The focus on household transmission in this paper necessitated including households into the modelled social structure; this represented significant additional effort in model construction, computational resource and inference compared to simpler models. A more common approach in the literature is to treat the contact rates between individuals as being determined only by their respective ages. This approach has the benefit of being conceptually straight-forward and draws on a number of recent and high-quality studies which quantify social contact patterns by age stratification (*Mossong et al., 2008*; *Kiti et al., 2014*; *Prem et al., 2017*). However, the fundamental theory of age-structured transmission models for endemic diseases was developed mainly with reference to diseases that induce very long term or lifelong immunity (*Anderson and May, 1991*). For diseases provoking long-lasting immunity, one would expect most older household members to be immune and therefore household structure to be a relatively less important factor in predicting risk of transmission compared to the age-structured transmission outside of the household. Indeed, simulation study of a generic strongly immunizing infection with realistic demography found limited difference in predicted incidence rate by age for people at schooling age or older between models with household structure and age structure compared to models with only age structure (*Geard et al., 2015*). However, it is not clear that neglecting household structure is a good approximation for modelling seasonal RSV transmission for two reasons: first, previously infected people lose effective immunological protection to RSV rapidly enough that each season could be closer to an 'epidemic' scenario rather than an 'endemic' scenario. Second, every hospital admission at KCH confirmed as due to RSV was a pre-school aged child; in contrast to predicted incidence rates for school age and older individual, the simulation study cited above (*Geard et al., 2015*) predicted that incidence was lower for 0–5 year olds, especially so for under 1 year olds, once household structure was taken into account. It would be of great interest to have a more general theoretical understanding of which epidemiological questions require household structure, or a more general meta-population structure, for epidemiological modelling, and which don't. This remains an active area of research (*Ball et al., 2015*).

A cocooning protective effect of households could explain the big discrepancy between our estimate of the mean period of protection against RSV after birth due to transplacental transfer of antibodies from mother to baby in the the womb (21.6 days of natural protection on average) compared to a RSV transmission modelling study by Kinyanjui et al on the same population using an age-structured model (*Kinyanjui et al., 2015*) (2.3 months of natural protection if the age mixing was based on diary estimates of contacts (*Kiti et al., 2014*) or 4 months of natural protection if the age mixing was based on household co-occupancy and schooling ages). The age-structured model used in the Kinyanjui et al study reported high or very high reproductive ratios: 7.08 for the diary based contact patterns, and 25.60 for the household co-occupancy and schooling age based contact pattern. Therefore, to fit the KCH hospitalisation data the age structured model necessarily predicted a very high level of natural protection due to maternal antibodies to compensate for the predicted high force of infection on young infants. In our model, we included household structure and we fit to the same KCH data but with a much lower level of natural protection from RSV. This in turn changes the guidance modelling gives to vaccination strategy; some age structured RSV transmission models have emphasized reducing force of infection by vaccinating infants directly (*Kinyanjui et al., 2015*), and find that maternal vaccination is likely to be of limited impact (*Pan-Ngum et al., 2017*), because they have inferred that the RSV reproductive ratio is high and, therefore, natural protection to RSV is also inferred to be high. In contrast, we infer that natural protection to RSV is low and therefore find that maternal vaccination in combination with elevating the cocoon protection to young infants provided by vaccinating household co-inhabitants is a highly efficient strategy. Another age-structured RSV transmission model (*Yamin et al., 2016*) has found that vaccinating under-fives to RSV along with their influenza vaccination was highly efficient because of the large number of secondary cases generated per infected under-five year old. Again, it is not clear whether this result extends to a

population structured into households where it is known that clustering in contacts has a complex interplay with disease dynamics, either reducing spread because infectious contacts are 'trapped' in the local cluster (e.g. the household) or promoting spread by enhancing persistence (*Miller, 2009*; *Sun et al., 2015*).

This was a modelling study and, as ever, there are factors that we have neglected in our analysis that could be addressed in future work. First, we treated coverage of the maternal vaccine and the IRP vaccine as independent. In reality, the simplest and cheapest scenario whereby the household cohabitants of pregnant mothers are recruited to the vaccination programme is if they attend prenatal contact with the mother-to-be. The percentage of pregnant women for have at least one prenatal contact in Kilifi county is high (97.5%; *KNBS, 2015*), however it is not clear that prenatal contact always occurs in the mother-to-be's third trimester. Both the MAB and IRP vaccines are likely to be best deployed late in the pregnancy, in order to maximise direct protection from the MAB vaccine and the duration of indirect protection from the IRP vaccine for the newborn. This means that if the only prenatal contact with the mother-to-be is relatively early in her pregnancy then both the MAB and IRP vaccines might fail; that is the households outside of MAB coverage are also likely to be those outside of IRP coverage violating our independent deployment assumption. Our results suggest that a MAB vaccine at a high coverage sharply reduces RSV hospitalisation even when the amount of additional protection is low (15 days) and if the MAB vaccination coverage is reduced to 50% IRP coverage becomes relatively more important to reducing hospitalisations. To avoid having many household unprotected by both MAB and IRP vaccination, it could be cost effective to devote extra resources towards encouraging pregnant women, and their cohabitants, who present early in the pregnancy to return for vaccination later in the pregnancy. Second, the cost per vaccine remains unknown and we have not considered any measurement of the burden of disease other than hospitalisations at KCH. RSV hospitalisations have been identified as a crude proxy for the true disease burden; the passive reporting of RSV hospitalisation can vary for reasons completely independent of RSV epidemiology (*Modjarrad et al., 2016*). Third, despite accounting for demographic change in our inference of model parameters we neglect demographic change in our forecasting, concentrating instead on predicting the reduction in hospitalisations compared to a baseline of a static population without intervention. Including demographic change in our parameter inference step allowed us to disentangle seasonal variation in hospitalisation from simply changing numbers of at-risk children. The demography in Kilifi will continue to change in the future, the crude birth rate in Kilifi has followed a declining trend in line with the rest of Kenya. However, this leads to a total birth rate which is much closer to static (~8500 births per year), and therefore the number of at-risk under-ones has been approximately static since 2009. We avoided exploring complications such as the effect increased crowding within households might have on the risk per-newborn in this paper by assuming that the rest of the population was also static over the 10 years of forecasting. Further exploring more detailed issues around shifting patterns of household cohabitancy would be an interesting avenue to explore in future work. Our primary goal in this paper has been to establish the importance of thinking jointly about hospitalisation risk, population structure (in particular household co-occupancy) and future vaccination programmes. We have demonstrated that, all other things be equal, combining partially effective vaccines can be complementary in a household-structured setting. These issues would suggest that RSV vaccination policy would benefit from further cost-benefit analyses tailored to LMIC settings, possibly using more flexible stochastic IBMs with the model parameters inferred in this study.

In conclusion, in this paper, we have analysed the performance of a joint maternal and household targeting RSV vaccination strategy measuring both reduction in hospitalisations and the true population incidence rate. We drew our conclusions based on rigorous inference of underlying transmission parameters and the inherent protection to RSV newborns received from their mothers, taking into account potential confusing factors such as variable seasonality and demography. Two central insights from our study were that the duration of natural protection to RSV that newborns inherit from their mother was likely to be much shorter than previously estimated and that RSV attack rates within the household were significant in maintaining RSV transmission. Therefore, targeting pregnant women and their households for RSV vaccination is likely to be an effective and efficient strategy under a wide range of different scenarios.

## Materials and methods

The dynamical RSV model used in this paper simulated infection and transmission of RSV among a population described by the Kilifi Demographic and Health surveillance system (KHDSS *Scott et al., 2012*) between September 2001 and September 2016. The population was assumed to mix and transmit RSV at two social levels: within their household and outside their household among the wider community. RSV infection was modelled using a modified version of the classic susceptible, infected, recovered (SIR) compartmental framework (*Anderson and May, 1991*; *Keeling and Rohani, 2008*). The main modifications were consistent with previous RSV transmission models; we assumed that: (i) individuals were born with a temporary immunity to RSV which faded over time, and (ii) RSV infection episodes provide individuals with only temporary protection from re-infection (mean 6 months *Scott et al., 2006*; *White et al., 2007*; *Moore et al., 2014*; *Pitzer et al., 2015*; *Kinyanjui et al., 2015*; *Yamin et al., 2016*). The high dimensionality of the ODE model (see below) used in this paper necessitated a relatively simple compartmental structure for RSV infection progression, therefore the population is only crudely age stratified into under-one year olds (U1s) and over-one year olds (O1s). However, more detailed information about the age of the individuals in the model was available by considering their age distributions conditional on their crude age category and the type of household they inhabited (see below). After an initial RSV infection there is evidence that individuals retain reduced susceptibility to subsequent RSV infection (*Henderson et al., 1979*; *Hall et al., 1991*), and will potentially have less infectious asymptomatic episodes if infected (*Hall et al., 2001*; *Yamin et al., 2016*). Some RSV transmission models, using simpler social structures, therefore allow individuals to be characterised by both their age and their number of previous RSV infections (*Kinyanjui et al., 2015*; *Yamin et al., 2016*). In the model used in this paper, we assumed that all U1 individuals susceptible to RSV were at risk of their first RSV episode and that all O1 individuals had already been infected at least once, since re-infection within the same yearly epidemic is unlikely but nearly everyone has caught RSV by the age of two years old (*Glezen et al., 1986*).

## Joint distributions of age and household occupancy

As mentioned above, the high dimensionality of the RSV transmission model with two levels of social mixing was a limiting factor on the possible complexity of the compartmental framework representing the possible combinations of age and disease state (see appendix 2). In order to both capture the structure of the population in households and incorporate finer-grained information about the ages of the modelled individuals, we calculated empirical joint distributions for the proportion of individuals of different ages in various household sizes, and whether that household contained an under-one year old. We did not restrict the age categories of this joint age-and-household distribution to just under-one or over-one, instead preferring finer-grained age categories: (i) each month of first year of life, (ii) each year of life aged 1–18 and (iii) 18+ years old. We used the Kilifi health and demographic surveillance system (KHDSS; *Scott et al., 2012*) to construct the joint distributions, which records for each individual a unique person ID, a birth date, immigration into the KDHSS date (s), out-migration from the KHDSS date(s), and a unique building ID for where they live during their time in the KHDSS. By combining this data we could calculate,

$$\mathbb{P}_t(a,n,U) = \frac{N_t(a,n,U)}{N_t}.$$

(1)

where $N_t(a,n,U)$ was the number of individuals on day $t$ who were jointly in age category $a$, lived in a household of size $n$, which either contained at least one under one year old ($U = 1$) or not ($U = 0$), and $N_t$ was the total population size on day $t$. The joint distribution changed over time, we calculated $\mathbb{P}_t(a,n,U)$ for a series of year-start days t = 1 st Jan 2000, 2001,..., 2016. We then used $\mathbb{P}_t$ as representative for the rest of the year. Because the exact birth dates where missing for a large number of people, and for model simplicity, we assumed that all U1 individuals aged to become O1 individuals at a constant rate 1 per year, which was equivalent to assuming that given that the exact age of an U1 individual was uniformly distributed between 0 and 1 years old, independently of the U1's household configuration.

## Conditional age of individuals

The dynamic model of transmission tracks whether individuals are under-one or over-one years old; however, for estimating the risk of disease per infection it was useful to use the conditional age distribution for the finer-grained age category of an individual based on her dynamic model age category $a<1\sim\mathrm{year}$ or $a>1\sim\mathrm{year}$, her household size and whether the household contained an U1 or not, for example,

$$\mathbb{P}_t(a|n,U,a>1\sim\mathrm{year}) = \frac{\mathbb{1}(a>1\sim\mathrm{year})\mathbb{P}_t(a,n,U)}{\sum_{b>1\sim\mathrm{year}}\mathbb{P}_t(b,n,U)}. \tag{2}$$

The conditional distributions for an individual's household size and whether they lived in a household containing an U1 based on their age were constructed similarly. The reason we included a variable indicating whether the household of the individual contained an under one or not was because it was important to capture the pathway to transmission to the under-one year olds most at risk of disease due to contracting RSV.

## Model dynamics, forces of infection and susceptibility to RSV

The fundamental unit of the RSV transmission model developed for this paper was the household. Each household was described by the number of each type of individual inhabiting it, which we call the *household configuration*. The type of individual within each household was identified by her RSV disease state and age category. The RSV transmission model described the dynamics of the number of households that were in each possible household configuration using an approach introduced by *House and Keeling, 2008*. Mathematically, the number of households in a given household configuration at time $t$ was denoted $H_{s_1,i_1,r_1,s_2,i_2,r_2}(t)$, referring to the household configuration with exactly $s_1$ U1 susceptibles, $i_1$ U1 infecteds, $r_1$ U1 recovered, $s_2$ O1 susceptibles, $i_2$ O1 infecteds, and $r_2$ O1 recovereds. In order to limit the number of possible household states, we included only households of total size ten or less with two or fewer under ones. We chose these limits on the household size based on capturing $\approx 99\%$ of the U1s in the population, and therefore the pathway to them catching RSV (appendix 2). There were 1926 possible household configurations in the RSV transmission model. The vector $H(t)$ of number of households in each possible household configuration evolved according to the semi-linear ODE:

$$\dot{H}(t) = A_t H(t) + f_t(H(t)) + \rho_t(H(t)). \tag{3}$$

Each term describing the vector field of *Equation (3)* corresponded to a dynamic component of the model:

1. RSV transmission within households, recovery of infected individuals, loss of immunity of recovered individuals, aging from U1 to O1 and turnover in household occupancy due to births and individuals leaving the household ($A_t H(t)$).
2. RSV transmission between households due to age-group specific mixing ($f_t(H(t))$).
3. Change in household numbers due to population flux, ($\rho_t(H(t))$).

See appendix 2 for further details. The force of infection due to transmission within a household of generic configuration $(s_1,i_1,r_1,s_2,i_2,r_2)$ was density dependent; that is the person-to-person infection rate in the household did not depend on household size,

$$\lambda_{hh} = \tau\beta(t)(i_1 + \iota_2 i_2). \tag{4}$$

where $\tau$ is the basic within-household transmission rate, $\iota_2$ is the relative decrease in infectiousness of O1s compared to U1s, and $\beta(t)$ is the seasonal variation in the transmission rate of RSV (see appendix 1). Transmission outside of the household within the wider community was assumed to be based on the finer-grained age categories introduced above. The conditional age distributions of the individuals allowed us to construct matrices ($P_{H\to A,t}$) to convert between the household configuration vector into a vector of number of infected individuals in each age category, weighted by their relative infectiousness, for any time $t$ during the simulation: $I(t) = P_{H\to A,t}H(t)$ (appendix 2). The force of infection on each individual due to age-based mixing in the community was,

$$\lambda_{age} = \beta(t)TI(t)/N(t). \tag{5}$$

where $T$ was the community infection rate matrix and $N(t)$ was the total population size at time $t$. In this formulation, the rate at which an infected in age group $b$ creates infectious contacts in the community with individuals of age group $a$ is $T_{ab}N(a,t)/N(t)$ where $N(a,t)$ is the number of individuals in age group $a$ at time $t$(Keeling and Rohani, 2008). The force of infection on an individual within a given household was calculated using matrices constructed from the conditional distribution of an individual's household type given her age, $\lambda_{com} = P_{A\rightarrow H,t}\lambda_{age}$. The total force of infection on each individual was the sum of her infectious contact rates within the household and within the community, $\lambda = \lambda_{hh} + \lambda_{com} + \lambda_{ext}$. Where $\lambda_{ext} = \epsilon\beta(t)/N(t)$ was the force of infection from outside KHDSS.

The actual infection rate for each individual was the force of infection 'felt' by the individual times the susceptibility of the individual. The susceptibility of under-one year olds ($\sigma_{U1}$) depended on whether or not the U1 individual was still protected from RSV by maternally acquired antibodies, which we modelled as giving a random $M$ days of protection; that is for an individual of age $A$ days, $\sigma_{U1} = 0$ if $M>A$ and $\sigma_{U1} = 1$ otherwise. In general, the infection status of an individual correlates with her age. However, because RSV is strongly seasonal we do not treat the age of an U1 as correlated with her susceptibility arguing that every U1 is facing her first RSV season irrespective of whether she is 1-month old or 11 months old. Therefore, the mean susceptibility for under-ones was $\overline{\sigma}_{U1} = \mathbb{P}(M \leq A)$. The susceptibility of over-one year olds was chosen as if the individual had definitely received at least one RSV infection in the past, and definitely had no chance of being maternally protected. We modelled the duration of maternal protection $M$ as a truncated exponential distribution conditioned on being less than 1 year in duration; that is $M \sim \exp(\alpha)|(M \leq 1 \sim \text{year})$ (appendix 2).

## Hospitalisation rates

The chance of an infected individual becoming severely diseased after contracting RSV, and then seeking care at hospital, depended on that person's age and number of infections (Nokes et al., 2008; Ohuma et al., 2012). When an U1 was infected in the model her age at infection was given by conditioning on the age of the U1 being greater than her maternal protection period,

$$\mathbb{P}(A \in a | M \leq A). \tag{6}$$

Which was calculated exactly (see appendices 2 and 4). This took into account that increasing the duration of maternal protection would increase the age at infection and therefore reduce the risk of disease. O1s were assumed to have no maternal protection but their conditional age depended on their household type [Equation (2)]. We used these conditional distributions to convert the incidence rate of U1s and O1s in each household type into dynamic incidence rates in each age category, $\mathcal{I}_a(t)$. By assuming that all O1s had been infected at least once we could use previously published age-dependent hospitalisation odds per infection $h_a$ (Kinyanjui et al., 2015 and appendix 3) to determine the cumulative hospitalisations predicted by the model for each age category $a$ and week interval $w_i = (t_{i,1}, t_{i,2})$,

$$\mathcal{H}(a, w_i) = \int_{t_{i,1}}^{t_{i,2}} \mathcal{I}_a(t)h_a dt. \tag{7}$$

## Parameter inference

The majority of the parameters for the RSV transmission model were drawn from the RSV literature (see Table 2 and appendix 3) leaving four parameters, and the five hyperparameters of a normal distribution describing the random yearly variation in log-seasonality, to be inferred from hospitalisation data (see Table 3 for parameter estimates and appendix 1 for further details on seasonality model). The free parameters and distribution of the RSV transmission model were:

1. Community infection rate outside the household between U1s and all others in the community accessing KCH ($b_{U1}$).
2. Community infection rate outside the household among all O1s in community ($b_{O1}$).
3. Infectious contact rate within the household to all other household members ($\tau$).
4. Mean duration of maternally derived immunity to RSV ($M$).
5. The joint normal distribution of the yearly log-seasonality amplitude and phase ($[\xi, \phi] \sim \mathcal{N}(\mu, \Sigma)$).

**Table 2.** Parameters from literature and chosen for model.

| Parameter | Description | Value | Data source |
|---|---|---|---|
| $\sigma_{O1}$ | Susceptibility (O1s) | 0.75 | *Henderson et al., 1979* |
| $\iota_2$ | relative infectiousness (O1s) | 0.5 | *Kinyanjui et al., 2015* |
| $\nu$ | Rate of waning of immunity | two per year | *Agoti et al., 2012* |
| $\gamma_1$ | Rate of recovery for under-ones | 1/9 per day | *Hall et al., 1976* |
| $\gamma_2$ | Rate of recovery for over-ones | 1/4 per day | *Hall et al., 1976* |
| $b_S$ | Community transmission rate at schools | 0,1/3,2/3,1 per day | range |
| $\eta$ | Ageing rate for U1s | one per year | model choice |
| $\epsilon$ | Base external infection rate (whole population) | 10 per day | model choice |

We also included an infectious contact rate for children of schooling age (5–18 years old; $b_S$) which acted additionally to $b_{O1}$; that is children of schooling age were at additional risk of contracting RSV on top of the risk due to mixing in the community. This meant that the mixing matrix in *Equation (5)* was in block form,

$$T = \begin{pmatrix} b_{U1} & b_{U1} & b_{U1} \\ \hline b_{U1} & b_S + b_{O1} & b_{O1} \\ \hline b_{U1} & b_{O1} & b_{O1} \end{pmatrix}. \tag{8}$$

where the blocks represented respectively under-one age categories, over-ones at school age categories and over-ones above school age categories. Unfortunately, we were unable to reliably identify $b_S$ parameter jointly with the other parameters. Investigating a range of $b_S$ values gave similar results for model fit and predictions for vaccine efficacy, the results in the main paper were for the highest value of $b_S$ considered which was mildly pessimistic compared to $b_S = 0$ (see appendix 3).

The data for parameter inference was RSV-confirmed, age-specific weekly admissions to Kilifi county hospital (KCH) hospitalisation data from September 2001 until September 2016 (see *Nokes et al., 2009* for study details). KCH serves as the primary care facility for the KHDSS population, and we assumed that all KHDSS members who accessed urgent hospital treatment due to RSV disease accessed their treatment at KCH. However, a significant number of admissions were from people not within the KHDSS survey leading to data re-scaling (see appendix 3). The log-likelihood for a particular simulation corresponded to Poisson errors,

$$\ln \mathcal{L} = \sum_i \sum_a \ln f_{poi}(\mathcal{D}_{i,a} | \mathcal{H}(a, w_i)). \tag{9}$$

**Table 3.** Inferred parameters.

| Parameter | Description | Value |
|---|---|---|
| $b_{U1}$ | Community transmission rate for U1s | 0.22 [0.18,0.27] per day |
| $b_{O1}$ | Community transmission rate for O1s | 0.20 [0.18,0.21] per day |
| $\tau$ | Transmission rate to *each* other member of household | 0.040 [0.032, 0.048] per day |
| $\overline{M}$ | Mean duration of maternal protection at birth | 21.6 [17.2, 26.1] days |
| $m_\xi$ | Mean amplitude of log-seasonality | 0.61 [0.51, 0.72] |
| $m_\phi$ | Mean timing of log-seasonality peak (phase) | 67.7 [40.2, 77.7] days |
| $\sigma_\xi$ | Std. amplitude of log-seasonality | 0.20 [0.098,0.31] |
| $\sigma_\phi$ | Std. timing of log-seasonality peak (phase) | 38.7 [30.0, 48.5] days |
| $\rho_{\xi\phi}$ | Corr. coefficient between log-seasonal amplitude and phase | −0.035 [-0.12, 0.072] |

where $\mathcal{D}_{i,a}$ was the cumulative number of hospitalisation observed at KCH in age category $a$ on week $w_i$ and $f_{poi}(x|\mu)$ is the probability mass function for a Poisson distribution with mean μ.

If the yearly realisations of the random seasonality (see appendix 1) were known, then the entire model would be deterministic and $\ln \mathcal{L}$ would be a function of the unknown parameters. Therefore, we treated the yearly variation in seasonality as missing data and used the Expectation-maximisation (EM) algorithm (*Dempster et al., 1977*) to converge onto maximum likelihood estimates for the four free parameters, and the two hyperparameters of the log-seasonality model, 95% confidence intervals were constructed using the likelihood profile technique (e.g. *King et al., 2008* and appendix 3).

## Modelling vaccination

There were two vaccines used in this modelling study, which were deployed as part of the prenatal contact between pregnant women and skilled health professionals. We assumed that the maternal vaccine was delivered as one injection to the pregnant women in her third trimester. This achieved some unknown additional period of maternal protection, $P$ days, on top of the random period $M$, that is after maternally vaccinating the period of protection became $M_{vac} = M + P$. Achieving an effective maternal vaccination coverage of $V_{cov}$ shifted the mean susceptibility of U1s to $\overline{\sigma}_{U1} = \mathbb{P}(M_{vac} < A)V_{cov} + \mathbb{P}(M < A)(1 - V_{cov})$, a linear increase in $V_{cov}$. The change in distribution of age at infection was non-linear in $V_{cov}$ because, conditional on an U1 being infected, it was more likely that the U1's mother had not been vaccinated than the unconditional probability of non-vaccination, $1 - V_{cov}$ (see appendix 4). We also assumed that there was a vaccine available that provoked an immune response in the vaccinated individuals similar to a natural infection; that is a susceptible $O1$ who is vaccinated immediately becomes 'recovered' and immune to RSV infection until her immunity waned. Immune response provoking vaccination was offered to all O1s in households when a birth occurred, as an addendum to the prenatal contact between mothers-to-be and health professionals. In principle, there were three dimensions to the coverage of the immunity provoking vaccine: (i) coverage of households, (ii) coverage within households, and (iii) vaccine effectiveness. For simplicity, we bundled these dimensions together, and vaccinated whole households at an effective vaccination coverage (the product of the three dimensions of coverage). Over 10 years of forecasted RSV epidemics if a MAB vaccine was available, and given to every pregnant mother, 8601 MAB vaccines were deployed each year. 0–24,095 IRP vaccines were deployed each year depending on household coverage. It should be noted that by 2016 the KHDSS population was around 240,000 people, hence 100% effective coverage of the households where births occurred corresponded to ~10% effective coverage of the total population.

## Model simulations

We simulated the model by numerically solving the high dimensional ODE [*Equation (3)*] simultaneously with the ongoing cumulative hospitalisations in each age category, $\dot{\mathcal{H}}_a = h_a \mathcal{I}_a(t)$, which allowed us to solve for the model predicted weekly hospitalisations [*Equation 7*]. The initial state of the model was unknown. We initialised the model by starting with a completely susceptible population with the population demography set to mimic that of the KHDSS on 1st Jan 2000. We then simulated RSV transmission for 10 years, with demographic rates (e.g. birth rates) chosen to match those of KHDSS in year 2000 and the seasonal amplitude and phase of $\ln \beta$ set to their latest mean estimate, in order to provide an initial state of the household model. Finally, we ran the model from 1st Jan 2000 until 1st September 2001. This provided the initial point for comparison to hospitalisation data. Numerical solutions were provided using the Sundials CVODE solver (*Cohen et al., 1996*) implemented within the DifferentialEquations package for Julia 0.6 (*Rackauckas and Nie, 2017*). For retrospective simulations comparing model predictions to data (*Figure 2*), we used the most probable values of the yearly seasonality. For forecast simulations, we generated 500 realisations of yearly seasonality over 10 years from the distribution inferred in model inference, this gave 500 predictions for the time series of future hospitalisations. We typically presented medians of these predictions (e.g. *Figure 4*). The code for the RSV household model used in this paper, and the data used for parameter inference, is available from https://github.com/SamuelBrand1/RSVHouseholdModel (*Brand, 2020*; copy archived at https://github.com/elifesciences-publications/RSVHouseholdModel).

## Acknowledgements

This work was funded by the Wellcome Trust (Grant ref 102975), and was published with permission of the Director of KEMRI. Kat Rock kindly supplied some clipart for plotting.

## Additional information

### Funding

| Funder | Grant reference number | Author |
| --- | --- | --- |
| The Wellcome Trust | 102975 | David James Nokes |

The funders had no role in study design, data collection and interpretation, or the decision to submit the work for publication.

### Author contributions

Samuel PC Brand, Conceptualization, Data curation, Software, Formal analysis, Validation, Investigation, Visualization, Methodology; Patrick Munywoki, Conceptualization; David Walumbe, Conceptualization, Data curation; Matthew J Keeling, Conceptualization, Formal analysis, Supervision, Validation, Investigation, Methodology; David James Nokes, Conceptualization, Resources, Formal analysis, Supervision, Funding acquisition, Validation, Investigation, Visualization, Methodology, Project administration

### Author ORCIDs

Samuel PC Brand (iD) https://orcid.org/0000-0003-0645-5367
Patrick Munywoki (iD) http://orcid.org/0000-0001-9419-7155
David James Nokes (iD) http://orcid.org/0000-0001-5426-1984

### Decision letter and Author response

Decision letter https://doi.org/10.7554/eLife.47003.sa1
Author response https://doi.org/10.7554/eLife.47003.sa2

## Additional files

### Supplementary files

• Transparent reporting form

### Data availability

All data generated or analysed during this study are included in the manuscript, supporting files or on the cited Github Repository. Source data files have been provided for Figures 2-6.

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

## Appendix 1

### Modelling seasonality in RSV transmission among KHDSS

RSV is a seasonal virus, in temperate climates the peak month for RSV incidence tends to be consistent year-on-year. Therefore, modelling approaches aimed at understanding RSV transmission in temperate climates have used an annually periodic deterministic function, with the timing of peak infectiousness of RSV being either a model parameter (*Yamin et al., 2016*) or itself a function of climatic variable to be fitted using regression methods (*Pitzer et al., 2015*).

The seasonal drivers of RSV transmission in the tropics are less clear (*Paynter, 2015*). At KCH the most common trough month for RSV hospitalisations was September, which lead us to define the RSV 'year' as September - September. The most common month for peak hospitalisation in each RSV year was January, however there was significant variation in peak month between RSV seasons with peaks occurring in each month November - April between 2002 and 2016 (*Appendix 1—figure 1*).

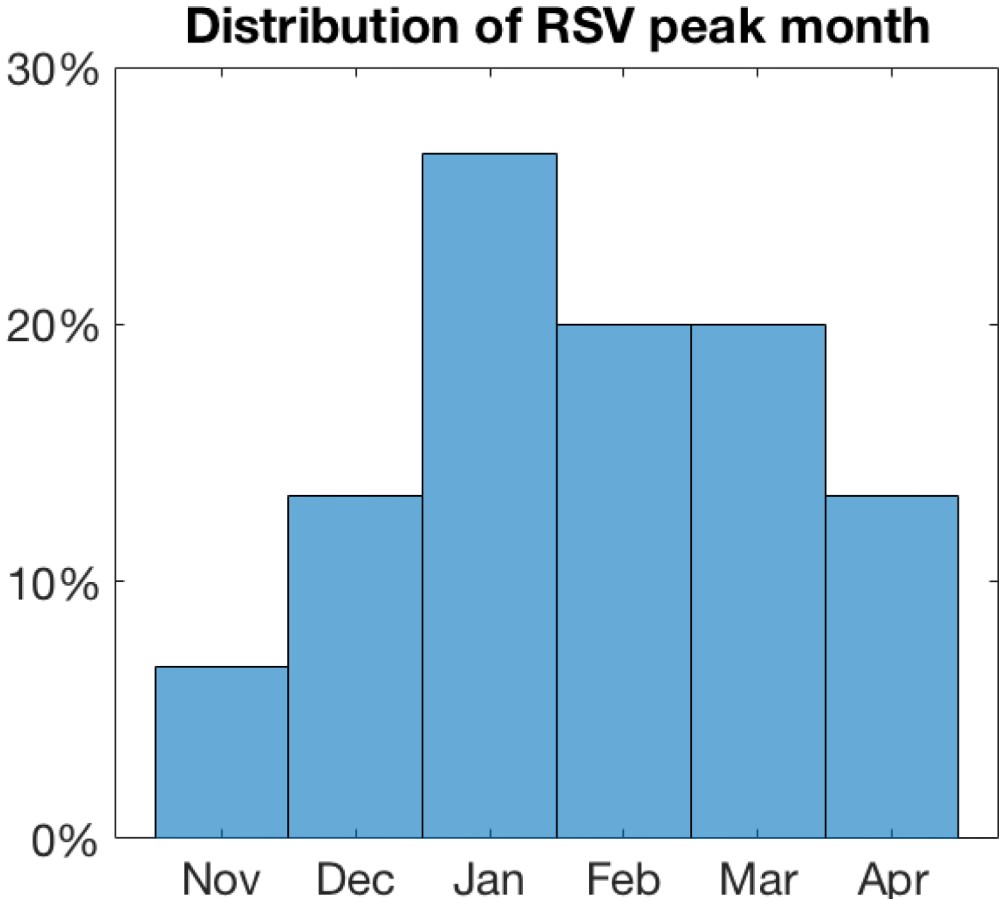

**Appendix 1—figure 1.** Distribution of peak month for RSV hospitalisations at KCH.

The year-on-year variation in peak month for RSV hospitalisation means that naively inferring a single fixed peak infectiousness parameter would not be a successful inference strategy. However, determining the precise mechanistic reason for shifting seasonality was challenging for the KHDSS population. RSV has been positively associated with the rainy season in some tropical settings (*Paynter et al., 2013*; *Paynter, 2015*); however, this is not obviously the case in Kilifi county where the rainy season is April to June with short rains October to December. There have been many proposed mechanisms for erratic periodicity in transmission (for a wide variety of infectious pathogens) which *could* be relevant to RSV transmission in Kilifi, for example, dynamical attractor switching (*Keeling et al., 2001*), or the effect of species/strain interaction (*Bhattacharyya et al., 2018*). In

particular, strain competition between RSV A and RSV B has been identified a mechanism for generating complex seasonal dynamics (*White et al., 2005*).

In this paper, we took an agnostic view and rather than choosing a mechanistic hypothesis for erratic seasonality from the many possible, we assume that the time-varying infectiousness of RSV alters randomly (but from a common distribution) year to year:

$$\ln \beta(t) = \xi_n \cos(2\pi(t - \phi_n)), \quad t \in \mathrm{RSV\,year}\,n. \tag{10}$$

where the RSV infectiousness ($\xi_n$) and seasonal peak timing ($\phi_n$) for each RSV year $n$ are drawn jointly from a normal distribution common to each year $(\xi_n, \phi_n) \sim \mathcal{N}(\mu, \Sigma)$. During model inference the yearly $\xi_n$ and $\phi_n$ realisations are treated as latent variables; their mean and covariance matrix are imputed along with other model parameters.

## Appendix 2

### Household- and age-structured RSV transmission model details

As described briefly in the main text, we developed a dynamic model for simulating the spread of RSV through the KHDSS population. The model was a hybrid between a mechanistic ODE approach, this included detailed household structure but only a simplified set of age-and-disease states for individuals within the households, and a data-driven empirical model, this used the observed joint distributions of KHDSS individuals' household occupancy and ages to generate conditional predications of individual detail beyond that of the mechanistic part of the model.

### Brief comparison to age-structured RSV transmission models

A commonly used conceptual framework for modelling epidemic transmission with a population is the compartmental model (*Anderson and May, 1991*; *Keeling and Rohani, 2008*); each person's disease state is described as being one of a finite number of possibilities, for example susceptible, infectious, recovered, which define that person's risk of contracting the infectious pathogen or transmissibility whilst infected with the pathogen. Additionally, it is usually important to capture the heterogeneity of the population, also called the *population structure*, in contrast to unstructured populations where every individual is treated as interchangeable. Therefore, each person will be described by their position in the population with sufficient detail that a rate of contact can be modelled between any pairs of individuals, see Diekmann and Heesterbeek for a more detailed discussion on modelling population structure (*Diekmann and Heesterbeek, 2000*). RSV transmission models have most commonly used age structure to describe heterogeneity in the population; each individual is described jointly by their disease state and which age interval (from some predetermined set of intervals) they occupy (*Pitzer et al., 2015*; *Kinyanjui et al., 2015*; *Yamin et al., 2016*). For age-structured RSV transmission models, there are two dynamical elements: the transmission of disease and the demographic turnover of the population (births, deaths and ageing). At the level of the individual these are modelled as discrete random events occurring at some per-capita rate (*Rock et al., 2014*). However, for large populations, there will be a very large number of individuals in each age-and-disease state, and the flux of population density in each age-and-disease state converges in probability onto the solution of a set of ordinary differential equations (ODEs) as the population size is treated as converging to infinite size (*Kurtz, 1970*; *Kurtz, 1971*; *Diekmann and Heesterbeek, 2000*). The limiting ODE model has as many degrees of freedom as there are age-and-disease state combinations in the epidemic model. In most epidemic modelling studies, it is the deterministic evolution of the solution to these ODEs that is usually given as the transmission model description.

In this paper, the essential modelling concept was to shift the focus away from numbers of individuals in each age-and-disease state and towards the number of households in each possible *household configuration*. A household configuration describes the number of individuals in each age-and-disease state who cohabit within a single household. Including households within the model adds a potentially relevant layer of realism; the social contacts within a household are *persistent*, therefore pairs of individuals that cohabit will repeatedly have the opportunity to infect one another if RSV enters the household but be relatively cocooned from infection if RSV has not entered the household. Age-structured transmission models implicitly assume that no two individuals contact one another more than once. To see this consider a population size of $N$; the rate of any individual contacting another single individual is $\mathcal{O}(1/N)$ therefore the probability that an individual selects the same other individual twice for contact over any finite time horizon goes to zero as $N \to \infty$ (which is also the limit at which the ODE model is valid). For household models the discrete random events that change the state of individuals (infection, death etc.) also change the household configuration. When the number of households is very large, there will be a large number of households in each possible household configuration and, as with age-structured models, there is convergence onto a set of ODEs with as many degrees of freedom as the number of possible household configurations.

The possible household configurations, or *state space*, of a household- and age-structured RSV transmission model is considerably larger than it would be for the equivalent age-structured model. If there are $m$ possible age-and-disease states then the number of possible household configurations for a household of size $n$ is given by a standard combinatorial identity, $\binom{n+m-1}{n}$. In this paper, we

consider a range of household sizes up to a maximum size $n_{max}$, therefore the number of household configurations was,

$$\#\text{household configurations} = \sum_{n=1}^{n_{max}} \binom{n+m-1}{n}.$$

The number of possible household configurations grows very rapidly (*Appendix 2—figure 1*). Therefore, having a sufficiently large $n_{max}$ to capture the target population required using a relatively simple compartmental age-and-disease state model for RSV infection.

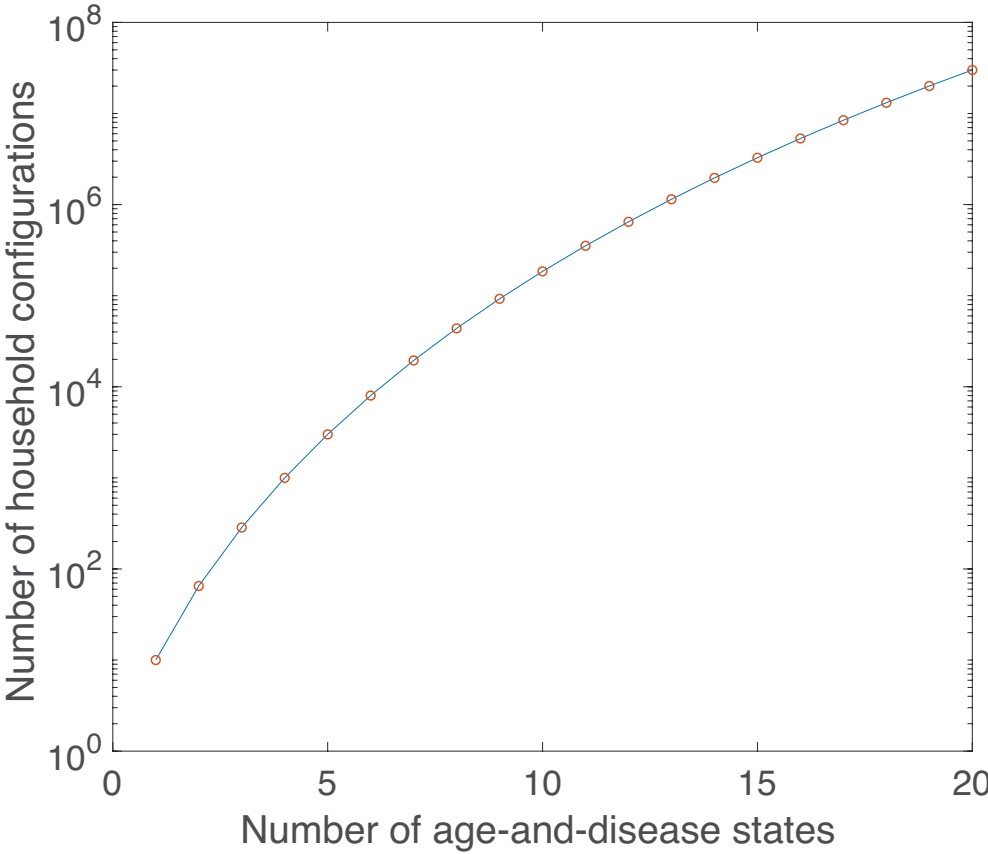

**Appendix 2—figure 1.** Growth in number of possible household configurations as complexity of the underlying age-and-disease state model grows. Calculated for a maximum household size of 10.

## Derivations for equilibrium behaviour of unstructured RSV transmission models

The age-and-household structured model we used in the main paper to make predictions of potential vaccine effectiveness in a population with persistent social structure. However, it can be useful to compare comparatively complex simulation studies to simpler models which are at least partially analytically tractable; this comparison identifies which features of a model are generic as opposed to emerging from more complicated factors (like seasonality or social structure).

A simple unstructured compartmental model of RSV transmission with two types of vaccine in a population of size $N$ was presented in the main paper (*Box 1*). Individuals are born into the population at rate $B$ and are initially protected against RSV by maternal antibodies ($M$). All individuals die at rate $\mu$. They lose maternal protection at rate $\alpha_{vac}$ (the rate associated with the maternal vaccine) and become susceptible to RSV infection ($S$). Each susceptible is infected at a rate $\beta I/N$ where $\beta$ is the product of the contact rate and the probability of transmission per contact and $I$ is the number of infected individuals in the population. Infected individuals clear their infection and become recovered and are temporarily immune to reinfection ($R$) at rate $\gamma$. Recovered individuals lose their

temporary immunity to reinfection at rate $\nu$. A vaccine aimed at provoking an immune response akin to a natural infection (IRP vaccine) is also used to control RSV. This is given to individuals in the population at effective rate $V$ (rate of delivery times probability the vaccine dose is successful). For simplicity, we assume that the IRP vaccine is not given to children so young they are likely to be in the $M$-compartment, but their isn't memory of which individuals have been vaccinated recently, therefore the chance that an individual selected for vaccination is actually susceptible is $S/(S+I+R)$. If a susceptible individual is vaccinated she transitions to becoming temporarily immune to RSV, this temporary immunity being lost at rate $\nu$.

The ODE equations for the dynamics of the basic unstructured model are:

$$\dot{M} = B - \alpha_{vac}M - \mu M, \tag{11}$$

$$\dot{S} = \alpha_{vac}M - \frac{\beta}{N}SI + \nu R - \mu S - V\frac{S}{S+I+R}, \tag{12}$$

$$\dot{I} = \frac{\beta}{N}SI - \gamma I - \mu I, \tag{13}$$

$$\dot{R} = \gamma I + V\frac{S}{S+I+R} - \mu R - \nu R. \tag{14}$$

We solve for the equilibrium state of this simple model, denoted $(M^*, S^*, I^*, R^*)$, assuming that the population has reached a steady size of $N$, with replacement birth rate $B = \mu N$. For the simple RSV model, we use a mortality rate $\mu$ that corresponds to a life expectancy of 65 years, the Kenyan average. The reproductive ratio for the model is $R_0 = \beta/(\gamma + \mu)$.

Since, the rate of loss of maternal immunity is fast compared to the mortality $(\alpha_{vac} \gg \mu)$ nearly all the population survive their $M$ period and become available for infection,

$$S^* + I^* + R^* = \frac{\alpha_{vac}}{\alpha_{vac} + \mu}N \approx N. \tag{15}$$

We use $S^* + I^* + R^* = N$ below to simplify the notation, but $N$ could be replaced with $N_{eff} = \frac{\alpha_{vac}}{\alpha_{vac} + \mu}N$. Note that the maternal vaccine does not alter the incidence rate for the simple RSV model at equilibrium, it simply delays the typical infection time. *Equation (13)* implies that either $I^* = 0$ (disease free state), or,

$$S^* = \frac{N}{R_0}. \tag{16}$$

Therefore,

$$R^* = N(1 - 1/R_0) - I^*. \tag{17}$$

Combining *Equations (12), (15), (16), (17)* gives that if RSV is endemic then,

$$I^* = \max\{\frac{1}{\gamma + \mu + \nu}((\mu + \nu)N(1 - 1/R_0) - V/R_0), 0\}. \tag{18}$$

*Equation (18)* implies that for the simple RSV model the critical rate at which an IRP vaccine eliminates RSV is $V_c = (\mu + \nu)N(R_0 - 1)$.

At an endemic equilibrium, the RSV incidence rate with vaccination rate $V$, denoted $\iota_V^*$, is therefore,

$$\iota_V^* = \frac{\beta S^* I^*}{N} = \frac{1}{(\gamma + \mu + \nu)R_0}((\mu + \nu)N(\beta - \mu\gamma) - (\gamma + \mu)V)$$
$$= \frac{(\gamma + \mu)}{(\gamma + \mu + \nu)R_0}((\mu + \nu)N(R_0 - 1) - V). \tag{19}$$

*Equation (19)* implies the two results which are presented in *Box 1* of the main text:

- The relative reduction in incidence due to IRP vaccination compared to no vaccination is,
  - 
$$\frac{\iota_0^* - \iota_V^*}{\iota_0^*} = \min\left\{\frac{V}{N(\mu+\nu)(R_0-1)}, 1\right\}. \tag{20}$$
  - In this paper, we model a scenario where co-habitants of newborn children each receive an IRP vaccine. This fixes $V$ to be proportional to the birth rate, $V = \mu N \langle H \rangle V_{cov}$, where $\langle H \rangle$ is the average number of co-habitants that a newborn has and $V_{cov}$ is the effective IRP coverage of households. This gives,
  - 
$$\text{Relative reduction in transmission due to vaccination} = \min\left\{\frac{\mu \langle H \rangle V_{cov}}{(\mu+\nu)(R_0-1)}, 1\right\}. \tag{21}$$

- Whilst RSV is not eliminated the reduction in incidence rate due to IRP vaccination is linear in $V$, with the improvement per extra vaccine used being a constant
  - 
$$\text{Reduction in transmission per IRP vaccine} = \frac{(\gamma+\mu)}{(\gamma+\mu+\nu)R_0}. \tag{22}$$

The mean number of over-one year olds living in households with at least one under-one year old in the KHDSS (see below) fluctuated yearly, but was never greater than five ($\langle H \rangle < 5$). Therefore, using a reversion to susceptibility rate $\nu = 2$ per year (see *Table 2*) with *Equation (21)* suggests that if, say, $R_0 = 2$ then the maximum achievable relative reduction in RSV incidence using this strategy with a Kilifi like population implied by the simple RSV model is 3.8%.

## Age-and-disease states for the household model

A literature review of mechanistic RSV transmission models revealed a number of critical common features:

- At birth newborns are born protected against RSV infection due to antibodies gained from their mother via trans-placental transfer. This is typically modelled as a maternally protected disease state $M$ e.g. (*Yamin et al., 2016*).
- The probability of developing severe disease and being hospitalised depends on a person's age, and number of times infected in the past, e.g. (*Kinyanjui et al., 2015*).
- The susceptibility to RSV infection per infectious contact, their infectiousness after infection, and the expected time taken to become recovered from RSV depend on number of times previously infected, e.g. (*Kinyanjui et al., 2015*).

The high dimensionality of household- and age-structured models necessitated using the most minimal age-and-disease state model possible for RSV (see above). To do this we use an extremely parsimonious approach. The possible age-and-disease state for individuals are: susceptible *or* maternally protected and under the age of one ($S_1$), infectious and under the age of one ($I_1$), recovered and under the age of one ($R_1$), susceptible and over the age of one ($S_2$), infectious and over the age of one ($I_2$) and recovered and over the age of one ($R_2$). An under-one year old (U1) experiencing some force of infection $\lambda$ becomes infected ($S_1 \rightarrow I_1$) and infectious to RSV at a rate $\sigma_{U1}\lambda$ where $\sigma_{U1}$ is the average susceptibility of an U1 year old to RSV. After becoming infected the U1 ceases to become infectious at a rate $\gamma_1$ ($I_1 \rightarrow R_1$) and then is immune to reinfection to RSV for a period of time. The immunity derived from natural infection is lost at a rate $\nu$, and the U1 revert to susceptibility but in the $S_2$ category ($R_1 \rightarrow S_2$). The reason we transition recovered U1s to a susceptible over-one year old (O1) is that due to the seasonality of RSV it is very rare for a person to be infected more than once in one epidemic season, therefore functionally by the time an individual is facing the risk of their second RSV lifetime infection they will very likely be over one. All U1s age at the rate $\eta = 1/365.25$ days$^{-1}$ becoming individuals in the same disease state but over-one ($S_1 \rightarrow S_2$, $I_1 \rightarrow I_2$, $R_1 \rightarrow R_2$). An O1 individual experiencing a force of infection $\lambda$ becomes infected and infectious ($S_2 \rightarrow I_2$) with RSV at a rate $\sigma_{O1}\lambda$ where $\sigma_{O1}$ is the relative susceptibility of O1s compared to an U1 no longer protected by maternal antibodies. Infectious O1s cease being infectious ($I_2 \rightarrow R_2$) at a

faster rate than U1s, $\gamma_2 > \gamma_1$, but revert to susceptibility ($R_2 \rightarrow S_2$) at the same rate $\nu$ (**Appendix 2— figure 2**).

As mentioned in the main document we relate this simple age-and-disease state model to more complicated RSV models by (i) using the conditional age distribution of individuals to address questions that required a more complicated age structure than a simple under/over-one binary choice, for example whether susceptible under ones were still protected by maternal antibodies, and (ii) by assuming that all over-ones have been infected at least once and all susceptible U1s have never been infected and *might* still be protected by maternal antibodies.

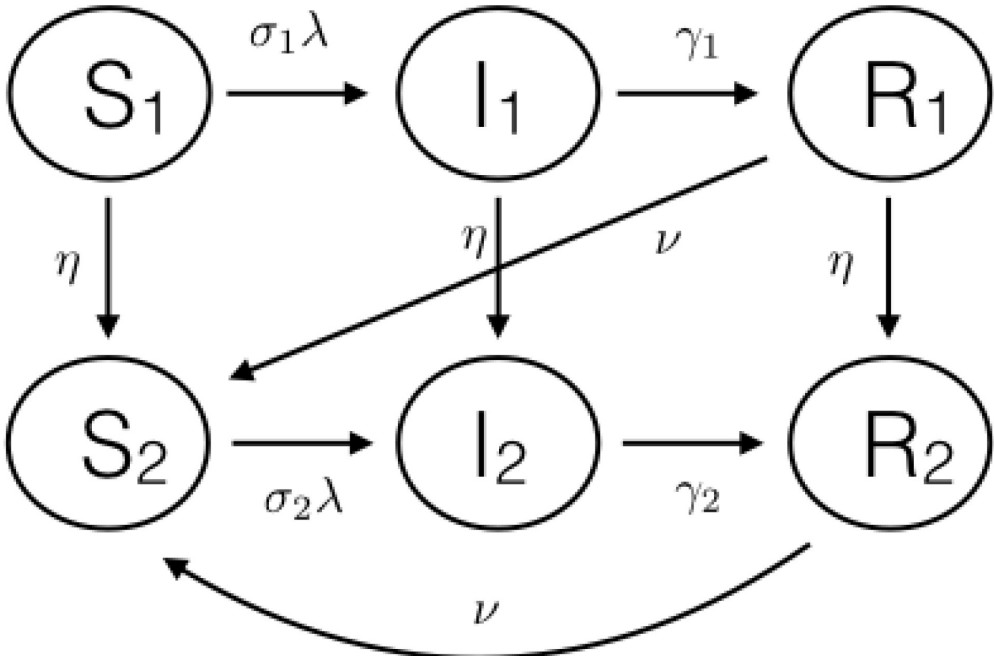

**Appendix 2—figure 2.** Schematic diagram of the basic age-and-disease state compartmental model for the individuals inside the households.

### Household- and age-structured model dynamics

A *household configuration* is a tuple of the number of individuals in each age-and-disease state who cohabit a household. The generic household configuration is denoted $h = (s_1, i_1, r_1, s_2, i_2, r_2)$, indicating that the household has precisely $s_1$ individuals in state $S_1$, $i_1$ individuals in state $I_1$ etc. The *household* size is the number of people living in the household (i.e. $s_1 + i_1 + r_1 + s_2 + i_2 + r_2$). We denote the space of possible household configurations $\Sigma$ and number of households in the state $h$ at time $t$ as $H_h(t)$. It is useful to consider a vector quantity over all possible household configurations such as $H(t) = (H_h(t) \sim | \sim h \in \Sigma)$ where we have generated some ordering for elements $h \in \Sigma$. It is clear that the knowledge of $(H(t), t \geq 0)$ would allow us to reconstruct the dynamics of individuals. For example, using the function $f(h) = s_1$ for each $h \in \Sigma$ in a vectorised form $f = (f(h) \sim | \sim h \in \Sigma)$ allows us to track the dynamics of numbers of individuals: $(f \cdot \mathbf{H}(t), t \geq 0)$.

As mentioned above, age-structured models are constructed by considering the per capita rate of events affecting the state of individuals. Household- and age-structured models are constructed by considering the per household rate of events that affect the household configuration (see *House and Keeling, 2009* for further mathematical details). In the following we list the events that change the household model divided into three groups: events due to transmission within the household, events due to transmission between households and events due to demographic turnover.

Events due to RSV transmission within the household

- Infection of susceptibles from within the household:

$$\text{For U1s}: \sim [s_1, i_1, r_1, s_2, i_2, r_2] \rightarrow [s_1 - 1, i_1 + 1, r_1, s_2, i_2, r_2] \sim \sim \text{at rate} : \sim \sigma_{U1}\beta(t)\tau s_1(i_1 + \iota_2 i_2), \quad (23)$$

$$\text{For O1s}: \sim [s_1, i_1, r_1, s_2, i_2, r_2] \rightarrow [s_1, i_1, r_1, s_2 - 1, i_2 + 1, r_2] \sim \sim \text{at rate} : \sim \sigma_{O1}\beta(t)\tau s_2(i_1 + \iota_2 i_2). \quad (24)$$

$\tau$ is the household infection rate, $\tau 2$ is the reduction in infectiousness due to being an O1, $\beta(t)$ is the seasonally varying component to the transmission rate and $\sigma_{O1}$ is the reduction in susceptibility due to being O1. Note that the true infection rate for U1s is $\sigma_{U1}\lambda_{hh}$ and for O1s is $\sigma_{O1}\lambda_{hh}$ as defined in main text. $\sigma_{U1}$ is the probability that an U1 individual is no longer protected by maternal antibodies, calculated by integrating over the individuals conditional age distribution as follows. Maternal protection was assumed to be 100% effective but only for a random duration per newborn of $M$ days, therefore using the uniform age distribution conditional on the individual being under one years old (see above),

$$\sigma_{U1} = \frac{1}{T}\int_0^T \mathbb{P}(M \leq a) \sim \mathrm{d}a. \quad (25)$$

where $T$ is the duration of a year expressed in the units of the simulation (we used days so $T = 365.25$ days). The probabilistic model for the duration of maternal protection was $P \sim \exp(\alpha)|M \leq T$ days , where $\alpha$ is the waning maternal immunity rate. The distribution function for $M$ is

$$\mathbb{P}(M \leq a) = \begin{cases} (1 - \exp(-a/\bar{M}))/(1 - \exp(-T/\bar{M})) & 0 \leq a \leq T \\ 1 & \text{otherwise} \end{cases} \quad (26)$$

where $\bar{M} = 1/\alpha$ is the mean period of maternal protection without conditioning on $M \leq T$, the true mean period of protection is $\mathbb{E}[M] = \bar{M} - T/(e^{T/\bar{M}} - 1)$ but this turns out to be a very small correction to $\bar{M}$ since we fit to $\bar{M}$ being less than 30 days (see below), therefore for simplicity we call $\bar{M}$ the mean duration of maternal protection to RSV. Substituting into *Equation (25)* and direct integration gives,

$$\sigma_{U1} = \frac{1}{1 - e^{-T/\bar{M}}} - \frac{\bar{M}}{T}. \quad (27)$$

Note that $\sigma_{U1} \approx 1 - \bar{M}/T$ when $\bar{M} \ll T$.

- Recovery of infecteds:

$$\text{For U1s}: \sim [s_1, i_1, r_1, s_2, i_2, r_2] \rightarrow [s_1, i_1 - 1, r_1 + 1, s_2, i_2, r_2] \sim \sim \text{at rate} : \sim \gamma_1 i_1, \quad (28)$$

$$\text{For O1s}: \sim [s_1, i_1, r_1, s_2, i_2, r_2] \rightarrow [s_1, i_1, r_1, s_2, i_2 - 1, r_2 + 1] \sim \sim \text{at rate} : \sim \gamma_2 i_2. \quad (29)$$

Where $\gamma_1$ and $\gamma_2$ are the recovery rates of U1s and O1s.

- Reversion to susceptibility:

$$\text{For U1s}: \sim [s_1, i_1, r_1, s_2, i_2, r_2] \rightarrow [s_1, i_1, r_1 - 1, s_2 + 1, i_2, r_2] \sim \sim \text{at rate} : \sim \nu r_1, \quad (30)$$

$$\text{For O1s}: \sim [s_1, i_1, r_1, s_2, i_2, r_2] \rightarrow [s_1, i_1, r_1, s_2 + 1, i_2, r_2 - 1] \sim \sim \text{at rate} : \sim \nu r_2. \quad (31)$$

- Where v is the reversion to susceptibility/waning immunity rate.

## Events due to RSV transmission from without the household

In a purely age-structured transmission model, the number of RSV infecteds in each age category, $I(t) = (I_a(t))_{a \in \mathcal{A}}$, is a dynamic model variable which evolves according to a set of ODEs. For the household- and age-structured model we derived $I(t)$ from the household configuration dynamics and the conditional age distributions as the expected number of infecteds in each category given the distribution of household configurations $H(t)$. Note that knowing a household configuration specifies both the household size $n = s_1 + i_1 + r_1 + s_2 + i_2 + r_2$ and the under-one occupant boolean $U = \mathbf{1}(s_1 + i_1 + r_1 > 0)$. Therefore, we could define a $|\mathcal{A}| \times |\Sigma|$ conversion matrix to convert between the dynamic $H(t)$ variables into the implied $I(t)$ variables,

$$P_{H \to A, t} = (\mathbb{P}_t(a|h))_{a \in \mathcal{A}, h \in \Sigma}, \tag{32}$$

$$I(t) = P_{H \to A, t} H(t). \tag{33}$$

The age-dependent force of infection on each individual in age category $a$, $\lambda_{age}(a)$ depends on a community age mixing matrix $T = (T(a,b))_{a \in \mathcal{A}, b \in \mathcal{A}}$,

$$\lambda_{age}(a,t) = \sum_{b \in \mathcal{A}} T(a,b)[\mathbf{1}(a < 1 \text{ year}) + \iota_2 \mathbf{1}(a > 1 \text{ year})]I_b(t)/N(t). \tag{34}$$

where $N(t)$ is the total population size at time $t$. This is a standard formulation for force of infection between different age groups (see *Keeling and Rohani, 2008*). In principle any age-mixing matrix can be used as $T$; however, we use a simple matrix in block form that differentiated only between U1s, O1s of school age, and all other O1s (see main text). The force of infection on U1 and O1 individuals within households was calculated using a $|\Sigma| \times |\mathcal{A}|$ conversion matrix, and a small force of infection from outside the KHDSS was added, $\epsilon$,

$$P_{A \to H, t} = (\mathbb{P}_t(h|a))_{h \in \Sigma, a \in \mathcal{A}}, \tag{35}$$

$$\lambda_{com}(U1, h, t) = \sum_{a < 1 \sim \text{year}} \mathbb{P}_t(h|a)\lambda_{age}(a,t) + \epsilon/N(t), \tag{36}$$

$$\lambda_{com}(O1, h, t) = \sum_{a > 1 \sim \text{year}} \mathbb{P}_t(h|a)\lambda_{age}(a,t) + \epsilon/N(t). \tag{37}$$

The external infection event changes the household configuration:
Infection of susceptibles from outside the household:

$$\text{For U1s:} \sim [s_1, i_1, r_1, s_2, i_2, r_2] \to [s_1 - 1, i_1 + 1, r_1, s_2, i_2, r_2] \sim \text{at rate:} \sim \sigma_{U1}\beta(t)s_1\lambda_{com}(U1, h, t), \tag{38}$$

$$\text{For O1s:} \sim [s_1, i_1, r_1, s_2, i_2, r_2] \to [s_1, i_1, r_1, s_2 - 1, i_2 + 1, r_2] \sim \text{at rate:} \sim \sigma_{O1}\beta(t)s_2\lambda_{com}(O1, h, t). \tag{39}$$

## Events due to demographic change in the population

In the household-and-age-structured RSV model, we track demographic change both by using the yearly updated joint distributions of age and household size and by the dynamics of the household configurations $H(t)$. The number of households of each size $n$ changed over time due to the effect of people leaving home, births, deaths, out-migration from KHDSS and in-migration into KHDSS. Moreover, the mean number of U1s per household of each size evolved over time. Rather than track all the possible events that change the demography of the KHDSS, we focus on (i) the ageing of the U1s becoming O1s, (ii) capturing the household size dependent birth rate, and (iii) capturing the change in household numbers for each household size.

The recorded birth rate that can be inferred from the KHDSS data set included newborns who out-migrate, neglected newborns that in-migrate at a very young age, and obviously some newborns die whilst very young. As mentioned above, we did not mechanistically track every possible demographic event, but instead calculated the *effective* birth rate that arrived at the correct mean number of U1s for each household size. For simplicity, we assumed that the effective birth rate was a

*turnover rate* for households; that is each birth is associated with a per-capita rate of an O1 leaving the household. This arrived at the correct density of U1s in the population, and in each size group of households, at the cost of assuming that events occurred at the same time rather than at the same rate.

The number of households of each size changed over time as the overall population size changed and individuals left households in order to form new households. As with the demographic turnover rate, there were multiple different mechanisms whereby new individuals entered the population and formed new houses or individuals and groups left the population, for example whole groups arrived and formed a new house, individuals arrived and joined houses etc. Moreover, the RSV infection status of the new entrants to the population were unknown. We assumed that new entrants arrived as households with the same distribution of household configurations as already observed in the population; that is that new arrivals didn't have a net effect on the *proportion* of individuals in each age-and-disease state just by arriving, although obviously as the population grew this has an effect of the number of hospitalisations we expected.

The demographic events that changed the household configurations were:
Aging:

$$[s_1, i_1, r_1, s_2, i_2, r_2] \rightarrow [s_1 - 1, i_1, r_1, s_2 + 1, i_2, r_2] \sim \sim \text{at rate} : \sim \eta s_1, \tag{40}$$

$$[s_1, i_1, r_1, s_2, i_2, r_2] \rightarrow [s_1, i_1 - 1, r_1, s_2, i_2 + 1, r_2] \sim \sim \text{at rate} : \sim \eta i_1, \tag{41}$$

$$[s_1, i_1, r_1, s_2, i_2, r_2] \rightarrow [s_1, i_1, r_1 - 1, s_2, i_2, r_2 + 1] \sim \sim \text{at rate} : \sim \eta r_1. \tag{42}$$

where $\eta = 1/T$ is the aging rate at which U1s become O1s. $T$ is the duration of a year expressed in the units of the simulation (we used days so $T = 365.25$ days).
Demographic turnover due to births and O1s leaving their household:

$$[s_1, i_1, r_1, s_2, i_2, r_2] \rightarrow [s_1 + 1, i_1, r_1, s_2 - 1, i_2, r_2] \sim \sim \text{at rate} : \sim \mu(n,t) s_2, \tag{43}$$

$$[s_1, i_1, r_1, s_2, i_2, r_2] \rightarrow [s_1 + 1, i_1, r_1, s_2, i_2 - 1, r_2] \sim \sim \text{at rate} : \sim \mu(n,t) i_2, \tag{44}$$

$$[s_1, i_1, r_1, s_2, i_2, r_2] \rightarrow [s_1 + 1, i_1, r_1, s_2, i_2, r_2 - 1] \sim \sim \text{at rate} : \sim \mu(n,t) r_2. \tag{45}$$

If there is at least one O1 left in the household, the birth/turnover rate is zero for households with only 1 O1; that is there are never any households of only U1s. $\mu(n,t)$ is the turnover rate per O1 household member in a household of size $n$ at time $t$ replacing them with susceptible U1s for households of size $n$. The turnover rates for each year were chosen so that the correct density of U1s per household was achieved (approximately). Following is a description of the fitting process so that the turnover rate lead to this household demography:

1. *Collect the empirical distribution of U1s per household size.* For each household size $n = 1, \ldots, n_{max}$ we calculated the mean number of U1s per household at y = 1st jan 2000-2017, this was denoted: $\overline{N}_{U1}(n, y)$.
2. Calculate the implied distribution of U1s per household size for any given birth/turnover rate. For any given birth/turnover rate, $\mu$, the equilibrium probability of finding $k$ U1s in a household of size $n$ is

$$\pi(k|n, \mu) \propto (\frac{\mu}{\eta})^k \binom{n}{k} \quad k = 0, \ldots, n-1, \tag{46}$$

$$\pi(n|n, \mu) = 0. \tag{47}$$

*Equation (46)* is just the equilibrium distribution of a birth-death process (*Grimmett and Stirzaker, 2001*).

3. *Matching the empirical distribution to the implied distribution.* We used a root-finder to find the turnover rate that matches the simulation's mean number of U1s per household of each size to the empirical data, *for the next year*:

$$\mu(n,t) \text{ is the solution to } \sum_{k=0}^{n-1} k\pi(k|n,\mu(n,t)) = \overline{N}_{U1}(n,y+1) \text{ for all } t \text{ in year } y. \quad (48)$$

Change in number of households due to population flux

$$[s_1,i_1,r_1,s_2,i_2,r_2] \to 2[s_1,i_1,r_1,s_2,i_2,r_2] \text{ at rate}: \frac{r(n,t)}{\sum_{h\in\Sigma_n} H_h(t)}, \text{ if } r(n,t) \geq 0, \quad (49)$$

$$[s_1,i_1,r_1,s_2,i_2,r_2] \to \emptyset \sim \sim \text{ at rate}: \frac{|r(n,t)|}{\sum_{h\in\Sigma_n} H_h(t)}, \text{ if } r(n,t)<0. \quad (50)$$

where $\Sigma_n = \{h = [s_1,i_1,r_1,s_2,i_2,r_2]\sim|\sim s_1+i_1+r_1+s_2+i_2+r_2 = n\}$ was the set of household configurations of households of size $n$. $r(n,t)$ was the daily rate of change of number of households of size $n$ interpolated between the empirical distribution dates.

## Simulating the model

The model above could in principle have an infinite number of states if the household size was not limited (see above). We chose limits on the household size based on capturing $\approx 99\%$ of the U1s in the population, and therefore the pathway to them catching RSV. The limits were: (i) no household is bigger than size 10, and (ii) no household has more than 2 U1s. This also covers the big majority of the total numbers of households (see *Appendix 2—figure 3*). The $n_{max} = 10$ limit was imposed by initialising the model without households of size >10, and setting $r(n,t) = 0$ for all $n$>10. The $\leq 2$ U1 limit was imposed by setting the birth/turnover rate to zero for all households with 2 U1s. Putting the limits in reduces the dimensionality of the system to 1926 different household configurations.

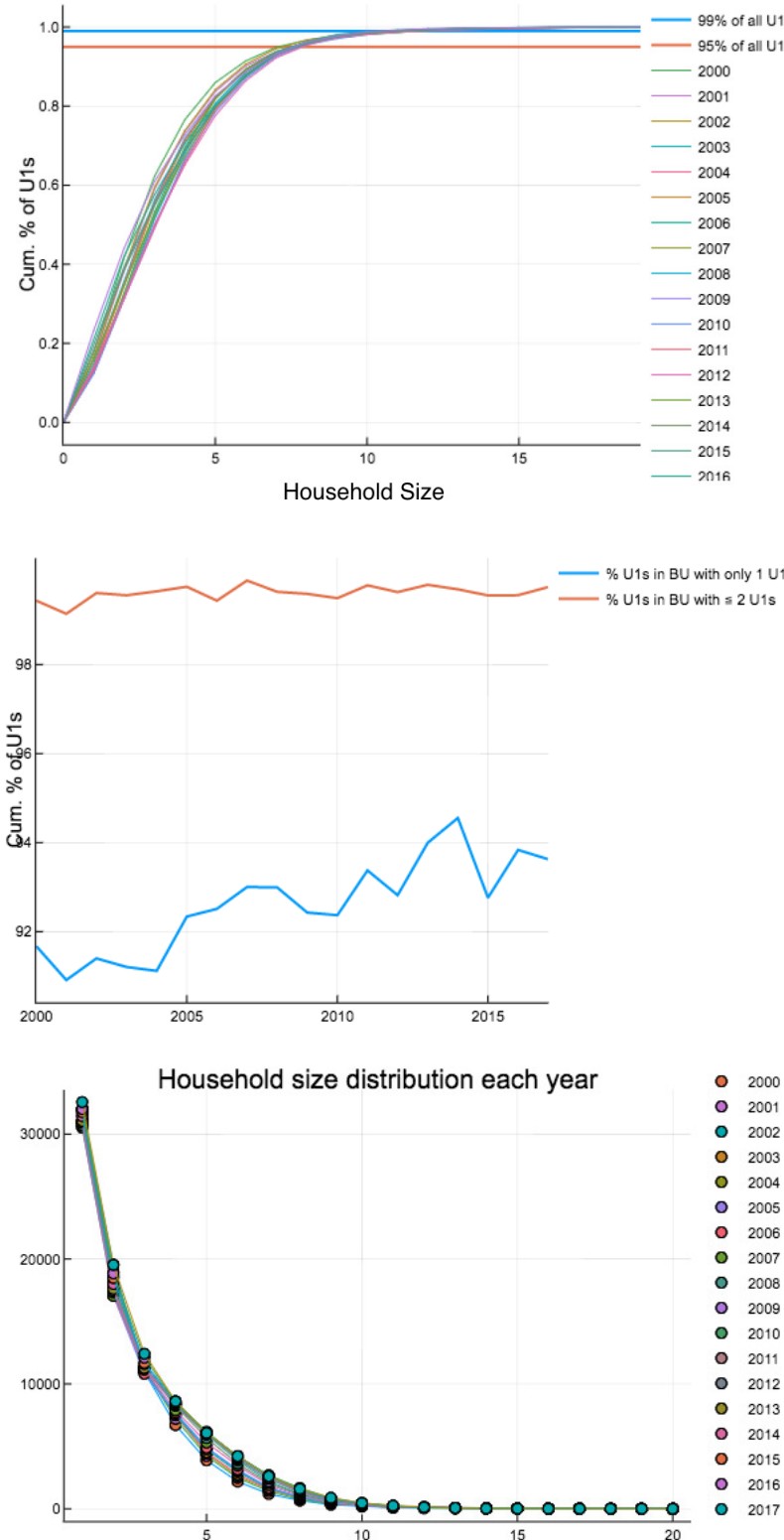

**Appendix 2—figure 3.** Household occupancy characteristics calculated on each 1 st Jan 2000–2017. *Top:* Percentage of U1s in households of a certain size or smaller. *Middle:* Percentage of U1s in households with only one U1 and households with one or two U1s. *Bottom:* Household size distribution.

Note that the events that either change a household's configuration or change the number of households described above can be divided into two categories: (*Nair et al., 2010*) those with rates that only depended on the household's configuration, e.g. infection within the household, or ageing of U1s, and, (*Glezen et al., 1986*) those with rates that depended on the configurations of other households, e.g. transmission between households or the rate of change of household numbers. The events in category (*Nair et al., 2010*) translate to linear dynamics for $H(t)$, events in category (*Glezen et al., 1986*) translate to non-linear dynamics (*House and Keeling, 2008*). Overall, the dynamics of $H(t)$ obey the semi-linear dynamical system,

$$\dot{H}(t) = A_t H(t) + f_t(H(t)) + \rho_t(H(t)). \tag{51}$$

$A_t$ is a matrix which encodes the dynamics of events in category (*Nair et al., 2010*), $f_t(H(t))$ encodes the transmission between households, and $\rho_t(H(t))$ encodes the rate of change of numbers of households in each configuration. We initialised the dynamics of *Equation (51)* by starting with a completely susceptible population on 1 st Jan 1990, allowing RSV to be introduced via the external force of infection and running for 10 years (see main text).

*Equation (51)* has two properties that are important to note:

- The change rate in households of size $n$ is independent of the transmission dynamics:

$$\partial_t \left( \sum_{h \in \Sigma_n} H_h(t) \right) = r(n,t), \qquad n = 1, \ldots, 10. \tag{52}$$

- The dynamics of the proportion of households in a given state $P_h(t) = H_h(t) / \sum_{h'} H_{h'}(t)$ is not directly affected by the change rates ($\rho_t$) in households:

$$\partial_t P_t = A_t P_t + \frac{f_t(H_t)}{\sum_{h'} H_{h'}(t)} \tag{53}$$

*Equations (52) and (53)* guarantee the desired modelling features discussed above. *Equation (52)* gives that the change in the number of households of each size matches the empirical rate of change for each year, we also verified this by numerical solution of *Equation (51)* (*Appendix 2— figure 4*). *Equation (53)* shows that the rate of change of household numbers doesn't directly effect the proportion of households in any given configuration. We also verified that the number of U1s and O1s was close to their empirical values (*Appendix 2—figure 5*).

*Equation (51)* was difficult to solve efficiently because it is both numerically stiff and high dimensional. We numerically solved *Equation (51)* using the Julia DifferentialEquations package implementation of the CVODE solver, with an efficient Krylov method (GMRES) to solve the implicit timestepping (see main text). We also used the DifferentialEquations efficient event handling which allowed us to change parameters (like the household change rate) at specific times without damaging the performance of the solver, or having to restart simulations.

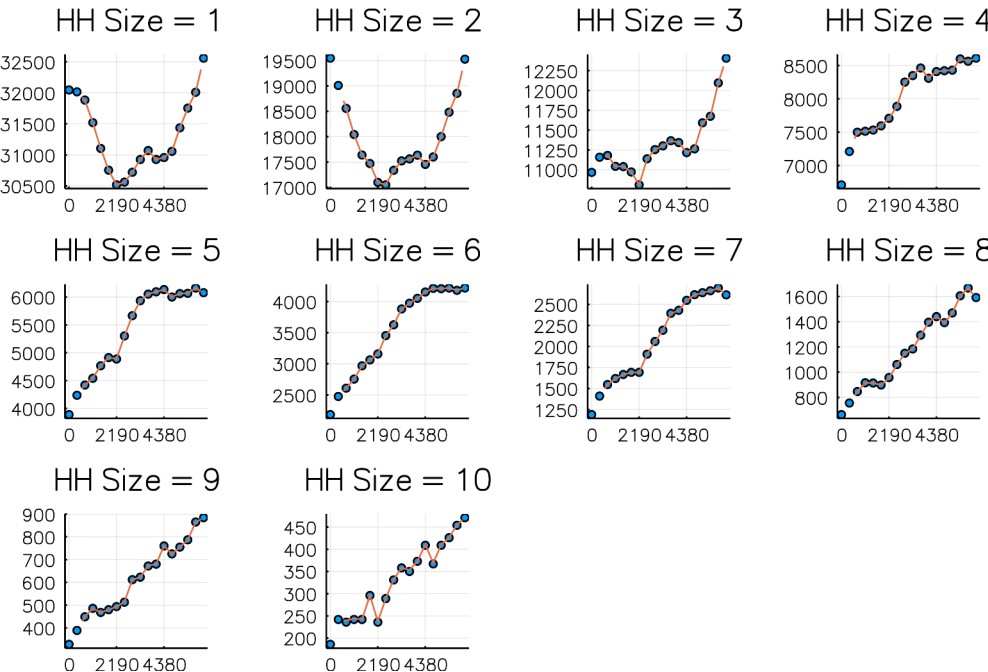

**Appendix 2—figure 4.** Comparison of numbers of households of sizes 1–10 on each 1 st Jan 2000–2017 (dots) against simulated values (curve). Simulation is from Sept 2001 - Sept 2016. Horizontal axis is days since 1 st Jan 2000.

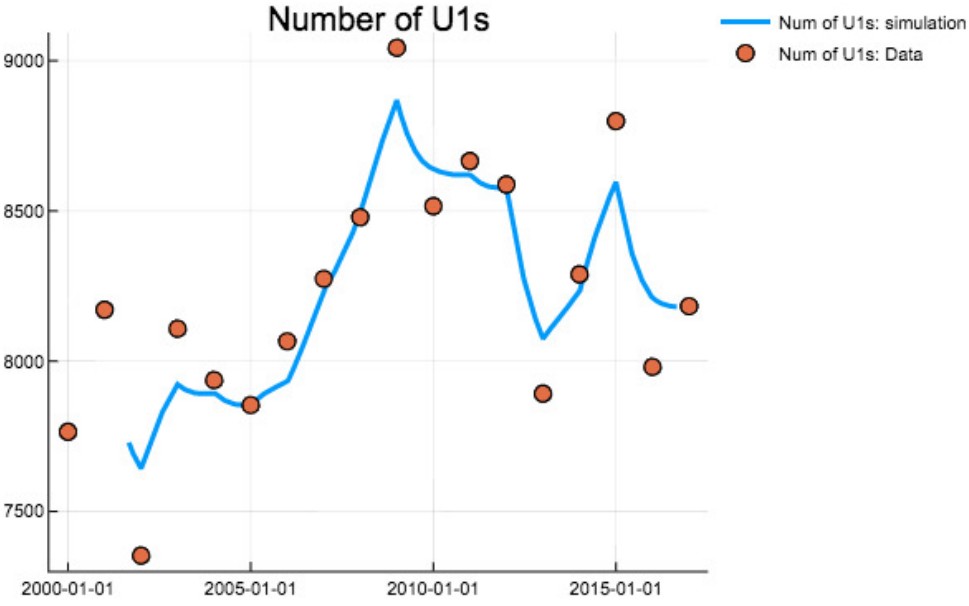

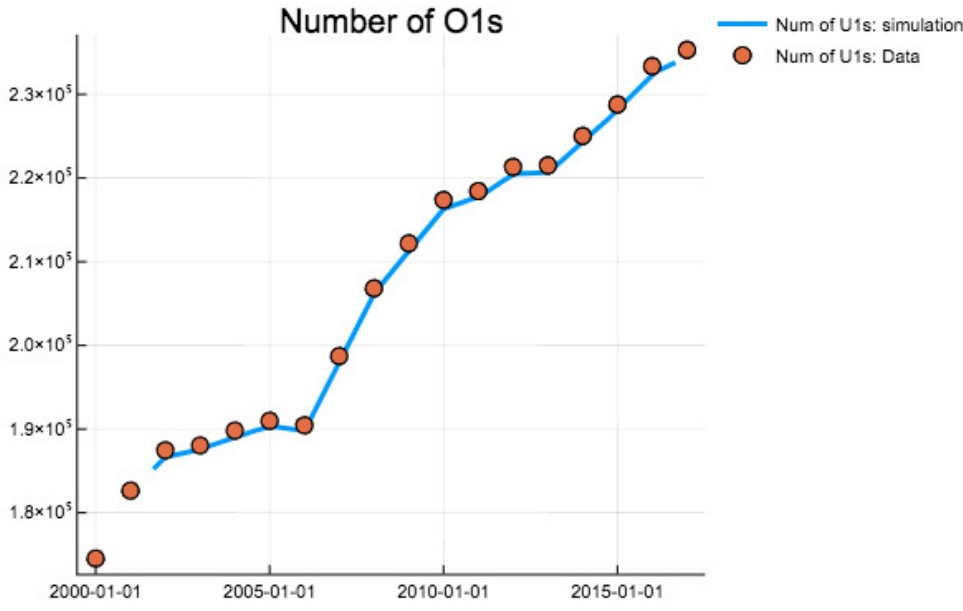

**Appendix 2—figure 5.** Comparison of total numbers of U1s and O1s on each 1 st Jan 2000–2017 (dots) against simulated values (curve).

# Appendix 3

## Parameters for the household- and age-structured RSV transmission model

The parameters for the household- and age-structured transmission model were drawn from four sources:

- A literature review of infectiousness duration and other epidemiological quantities; main *Table 2*.
- Calculated from the empirical joint distributions (see above *Appendix 3—table 1*).
- Age-dependent hospitalisation probability per RSV infection derived from *Kinyanjui et al., 2015*; *Appendix 3—table 2*. Hospitalisation probability was the probability that an infected individual would develop severe disease, multiplied by the probability that severely diseased individuals would require hospitalisation. The probability that an infected individual became diseased depended on whether it was the individual's primary infection episode or not. The underlying data for estimating these probabilities was drawn from cohort studies on RSV disease rates (*Ohuma et al., 2012*; *Nokes et al., 2008*). We adapted these probabilities for our model using our assumption that all infected under-ones were experiencing their first RSV episode, and all over-ones were experiencing their second or subsequent infection.
- Inferred from the KCH hospitalisation data set (see below).

**Appendix 3—table 1.** Parameters estimated from KHDSS data.

| Parameter | Description | Value | Data source |
|---|---|---|---|
| $\mu(n, t)$ | Birth/turnover rate for households of size n on day t | Varies, see above | KHDSS |
| $r(n, t)$ | Rate of change of numbers of households of size n on day t | Varies, see above | KHDSS |
| $P_{H \to A,t}$ | Conditional age distribution given household config. on day t | Varies, see above | KHDSS |
| $P_{A \to H,t}$ | Conditional household config. distribution given age category on day t | Varies, see above | KHDSS |

**Appendix 3—table 2.** Age-dependent hospitalisation probabilities per infection derived from *Kinyanjui et al., 2015*.

| Age category | Probability of hospitalisation per infection |
|---|---|
| 0-1 month | 0.10 |
| 1-2 month | 0.10 |
| 2-3 month | 0.063 |
| 3-4 month | 0.059 |
| 4-5 month | 0.054 |
| 5-6 month | 0.025 |
| 6-7 month | 0.019 |
| 7-8 month | 0.022 |
| 8-9 month | 0.012 |
| 9-10 month | 0.016 |
| 10-11 month | 0.013 |
| 11-12 month | $5.1 \times 10^{-3}$ |
| 1-2 years old | $2.6 \times 10^{-3}$ |
| 2-3 years old | $7.5 \times 10^{-4}$ |
| 3-4 years old | $2.2 \times 10^{-4}$ |
| 4-5 years old | $3.8 \times 10^{-5}$ |

## Parameter inference for the household- and age- model

As mentioned in the main text we used the EM algorithm (*Dempster et al., 1977*) to estimate parameters for the model. Again, as described in the main text the parameters we chose for inference were:

- Infectious contact rate outside the household between U1s and all others in the community accessing KCH ($b_{U1}$).
- Infectious contact rate outside the household among all O1s in community ($b_{O1}$).
- Infectious contact rate within the household ($\tau$).
- Rate of loss of maternally derived immunity to RSV ($\alpha$).
- The joint normal distribution of the yearly log-seasonality amplitude and phase ($[\xi, \phi] \sim \mathcal{N}(\mu, \Sigma)$).

where the community age mixing matrix $T(a, b)$ was in block form:

$$T = \begin{pmatrix} b_{U1} & b_{U1} & b_{U1} \\ b_{U1} & b_S + b_{O1} & b_{O1} \\ b_{U1} & b_{O1} & b_{O1} \end{pmatrix}. \tag{54}$$

The log-likelihood for our model [*Equation (8)* main text] was defined using the incidence rates $\mathcal{I}_a(t)$ predicted by solving the model. The incidence rate for all the households in the generic household configuration was,

$$\text{For U1s}: \sim \mathcal{I}_h(\text{U1}, t) = (\sigma_{U1} \beta(t) s_1 (\lambda_{hh} + \lambda_{com}(\text{U1}, h, t))) H_h(t) \tag{55}$$

$$\text{For O1s}: \sim \mathcal{I}_h(\text{O1}, t) = (\sigma_{O1} \beta(t) s_2 (\lambda_{hh} + \lambda_{com}(\text{O1}, h, t))) H_h(t). \tag{56}$$

where the household force of infection for the generic household configuration was $\lambda_{hh} = \tau(i_1 + \iota_2 i_2)$. We converted the household incidence rate into an age structured incidence rate by using conditional age distributions, and this allowed us to calculate the cumulative hospitalisations in age category $a$, predicted by a given set of parameters and yearly seasonality realisations, in weekly intervals $w_i = (t_{i,1}, t_{i,2})$ using the age-dependent hospitalisation rates per infection $h_a$ (see *Table 2*)

$$\mathcal{I}_a(t) = \sum_{h \in \Sigma} \mathbb{P}(A \in a | M < A, A \le 1 \text{ year}) \mathcal{I}_h(O1, t) + \mathbb{P}(a | h, A > 1 \text{ year}) \mathcal{I}_h(O1, t) \tag{57}$$

$$\mathcal{H}(a, w_i) = K(t) \int_{t_{i,1}}^{t_{i,2}} \mathcal{I}_a(t) h_a \sim dt, \tag{58}$$

$$\ln \mathbb{P}(\mathcal{D}_{i,a} | \theta, \xi, \phi) = l(\theta, \xi, \phi) = \sum_i \sum_a \ln f_{poi}(\mathcal{D}_{i,a} | \mathcal{H}(a, w_i)). \tag{59}$$

Here, $K(t)$ is a time-varying scale factor that accounted for the fact that whilst we were modelling RSV infection for the KHDSS population, other individuals were accessing KCH for treatment of RSV-induced severe disease. To fit $K(t)$, we first performed a polynomial regression $R(t)$ against the ratio of KHDSS members using KCH against non-KHDSS members (*Appendix 3—figure 1*) t = 0 (days) is 22nd April 2002 fitted curve is R(t) = 1.24+ 0.00224 $t$ - 2.45e-6$t^2$+ 9.45e-10$t^3$ - 1.55e-13$t^4$ + 9.10e-18$t^5$ for $t<0$, and $R(t)$ took its final value for times after 1st sept 2016. Having fitted the ratio, the scale factor was $K(t) = (1 + R(t))/R(t)$, which we derived by assuming that non-residents were experiencing RSV hospitalisations at proportionally the same rate as residents.

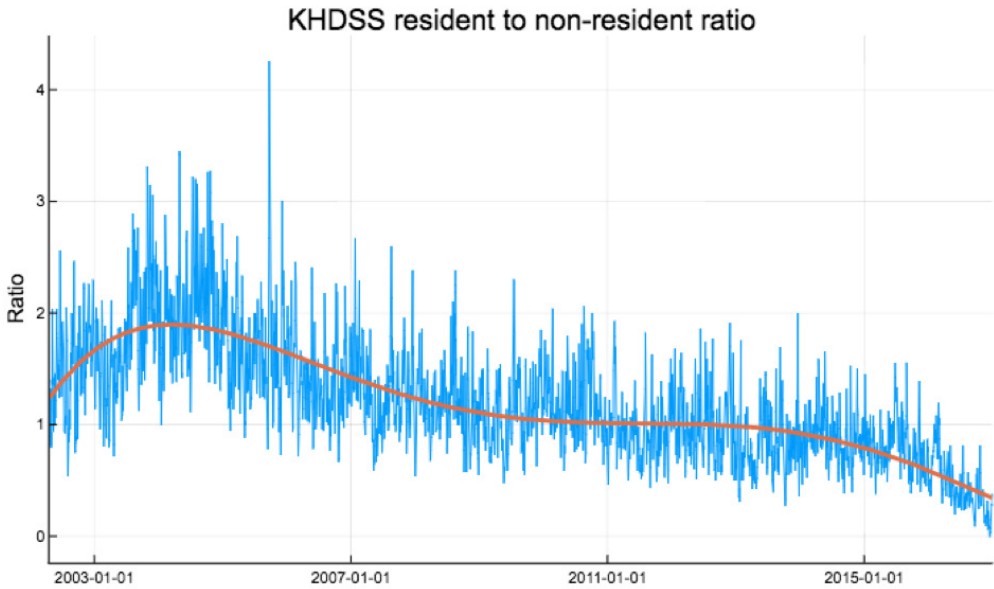

**Appendix 3—figure 1.** Ratio of KHDSS residents to non-residents weekly accessing KCH for confirmed RSV treatment. Red curve is polynomial fit $R(t)$.

The conditional age category of an U1 who has definitely been infected, where $a = (a_0, a_1)$,

$$
\begin{aligned}
\mathbb{P}(A \in a | M{<}A, A \leq 1 \text{ year}) \;&= 1(a \leq 1 \text{ year})\frac{\mathbb{P}(M{<}A|A \in a)\mathbb{P}(A \in a|a \leq 1 \text{ year})}{\mathbb{P}(M{<}A|a \leq 1 \text{ year})} \\
&= 1(a \leq 1 \text{ year})\frac{a_1 - a_0 + \overline{M}(e^{-a_1/\overline{M}} - e^{-a_0/\overline{M}})}{T(1 - e^{-T/\overline{M}})\sigma_{U1}}
\end{aligned}
\tag{60}
$$

An implication of expression (*Equation (60)*) is that if $a_0$ and $a_1$ are both significantly less than $\overline{M} = 1/\alpha$ then $\mathbb{P}(A \in a|M{<}A, A \leq 1 \text{ year}) \approx 0$; that is that, although we have assumed that the conditional age of an U1 is distributed evenly over the first year of life, the conditional age distribution of an U1 who has been infected is typically older than $\overline{M}$. This allowed us to extract information for inferring $\alpha$ from the age distribution of hospitalised children at KCH despite only using a crude U1/O1 age distinction in the mechanistic formulation of the household-and-age model. The log-likelihood $l(\theta, \xi, \phi)$ [*Equation(59)*] could be determined for a given set of parameters and realisations of the yearly seasonal amplitude and phase by solving the full ODE system numerically [*Equation (51)*], and thereby also calculating the weekly hospitalisations. $\theta$ represented the model parameters to be inferred, $\xi$ and $\phi$ were the vectors of the seasonal transmission model *Equation (10)*, and $\mathcal{D}_i, a$ was the KCH hospitalisation data for the ith week in the $a$ age category.

The main difficulty in the inference for the unknown parameters $\theta$ was that the actual realisations of $\xi$ and $\phi$ are not observed, therefore $l(\theta, \xi, \phi)$ could not be calculated directly. Instead, we use the EM algorithm to converge onto a maximiser of the marginal likelihood, $\mathcal{L}(\theta) = \int \mathbb{P}(\mathcal{D}, \xi, \phi|\theta) \sim \mathrm{d}\xi \sim \mathrm{d}\phi$. The EM algorithm converges a sequence of parameter estimates $(\theta^{(n)})_{n \geq 0}$ towards a local maximum of the marginal likelihood by alternatively, (1) calculating the expected value of the log-likelihood over the conditional distribution of $\xi$ and $\phi$ given the observed data $\mathcal{D}$ and the current estimate of the parameters, which we dub the $Q$ function [E step], and, (2) finding the parameters which maximised the $Q$ function [M step]. We now give details of how this was implemented for the specific model developed in this paper:

E step: The conditional distribution of $\xi$ and $\phi$ given the $n$-th parameter estimate $\theta^{(n)}$, from the previous M-step, and $\mathcal{D}$ could not be calculated in closed form. In principle, this distribution could have estimated numerically (e.g. by using a particle filter method), however, because the household-and age-structured RSV transmission model was comparatively slow to integrate (~ 40 secs per simulation) we resorted to saddle-point integration. Our argument is that because nearly every year has a

sharply peaked hospitalisation rate then, given a parameter estimate $\theta^{(n)}$, the conditional probability of $(\xi, \phi)$ should be concentrated around a particular value, making saddle-point integration an appropriate approximation (see [*Hinch, 1991*] for further details on saddle-point integration). Using the saddle-point approximation, we could solve for the $Q$ function,

$$
\begin{aligned}
Q(\theta|\theta^{(n)}) &= \mathbb{E}_{\xi,\phi|\mathcal{D},\theta^{(n)}}[\ln \mathbb{P}(\mathcal{D}, \xi, \phi|\theta)] \\
&= \mathbb{E}_{\xi,\phi|\mathcal{D},\theta^{(n)}}[l(\theta, \xi, \phi) + \ln \mathbb{P}(\xi, \phi|\theta)] \\
&\approx l(\theta, \xi^*, \phi^*) + \ln \mathbb{P}(\xi^*, \phi^*|\theta) \\
&= l(\theta, \xi^*, \phi^*) - \sum_i [(\xi_i^* - m_\xi) \sim (\phi_i^* - m_\phi)] \Sigma_{\xi\phi}^{-1} [(\xi_i^* - m_\xi) \sim (\phi_i^* - m_\phi)]^T + \mathrm{const.}
\end{aligned}
\tag{61}
$$

The approximation step in *Equation (61)* is the saddle-point integration approximation of the average, and the quadratic form is due to our assumption that the seasonal amplitude and phases are distributed jointly normally. Saddle-point integration is equivalent to assuming that the full mass of the conditional distribution of $(\xi, \phi)$ was concentrated at its most probable value,

$$
\begin{aligned}
(\xi^*, \phi^*) &= \arg\max_{\xi,\phi} \ln \mathbb{P}(\xi, \phi|\mathcal{D}, \theta^{(n)}) \\
&= \arg\max_{\xi,\phi} \{\ln \mathbb{P}(\mathcal{D}|\xi, \phi, \theta^{(n)}) + \ln \mathbb{P}(\xi, \phi|\theta^{(n)})\} \\
&= \arg\max_{\xi,\phi} \{l(\theta^{(n)}, \xi, \phi) - \sum_i [(\xi_i^* - m_\xi^{(n)}) \sim (\phi_i^* - m_\phi^{(n)})] \Sigma_{\xi\phi}^{-1,(n)} [(\xi_i^* - m_\xi^{(n)}) \sim (\phi_i^* - m_\phi^{(n)})]^T\}.
\end{aligned}
\tag{62}
$$

We determined $(\xi^*, \phi^*)$ by sequentially optimising *Equation (62)* over each season by simulating the model repeated and using the Nelder-Mead algorithm implemented within the Optim package for Julia 0.6. Note that saddle point integration has converted solving for the function $Q$ into a regularised maximum likelihood problem where the regularisation was provided by the mean and covariance matrix for log-seasonal amplitude and phase derived in the previous M step.

M step: Having constructed the $Q$ function associated with the $n$-th parameter iteration [*Equation (61)*], we maximised $Q$ over $\theta$. The maximum point of $Q$ being $\theta^{(n+1)}$ for the next E-step. Maximisation proceeded in three stages:

The maximising values for the mean and covariance matrix of the random seasonal amplitude and phase were given by maximum likelihood using $(\xi^*, \phi^*)$ derived in the E-step. This was performed using the fit_mle function provided by the Julia Distributions package.

We performed a global optimisation for $Q$ over a box in parameter space defined by limits $[0, 1]$ for transmission parameters and $1/\alpha = \overline{M} \in [10, 120]$ days for the inverse rate of loss of maternal immunity. Global optimisation was performed by running 600 iterations of a differential evolution optimiser (*Storn and Price, 1997*) with 50 agents. The differential evolution optimiser was implemented by the adaptive_de_rand_1_bin_radiuslimited optimiser from the Julia BlackBoxOptim package. The purpose of the global optimisation step was to reduce the dependence on choosing an initial guess about $\theta$ since the whole plausibility space of the parameters was explored at each iteration of the EM algorithm. We called the best performing agent's parameter set on the $(n+1)$ th step, $\tilde{\theta}^{(n+1)}$.

We used $\tilde{\theta}^{(n+1)}$ as the starting point for a further local optimisation of $Q$ using the Nelder-Mead algorithm implemented by the Julia Optim package. This step provided $\theta^{(n+1)}$ for the next E-step.

We iterated EM algorithm until no further improvement in the value of $Q^* = \max_\theta Q$ was achieved, and then retained $\theta^* = \arg\max_\theta Q$ as the maximum likelihood estimator for the parameters. 95% confidence intervals were estimated by using univariate profile likelihood for $Q$; that is varying one parameter at a time whilst keeping others fixed until a $\chi^2$ region was determined around the maximum of $Q$ (see *King et al., 2008* for a description of 95% CIs for dynamical systems).

## School mixing scenarios and inference results

We were unable to identify a mixing rate within schools $b_S$, see *Equation (54)*, therefore we considered four values of $b_S$ each determined by what a baseline reproductive value for RSV would be *if* only school children mixed together and the seasonality was just $\beta(t) = 1$, $R_S$, using the simple formula,

$$R_S = \frac{b_S \sigma_{O1} \iota_2}{\gamma_2} \tag{63}$$

These four scenarios were: zero schools transmission ($R_S = 0$), low schools transmission ($R_S = 0.5$), medium schools transmission ($R_S = 1$), and, high schools transmission ($R_S = 1.5$). We saw that once maximum likelihood estimation was performed on the free parameters: $\theta = (b_{U1}, b_{O1}, \tau, \alpha, m, \Sigma_{\xi\phi})$ the resultant fits to the data were very similar visually (see *Appendix 3—figure 2*). We noticed that the outcomes of vaccination were also similar for each four scenarios (see below and *Figure 1*). Therefore, for robustness of conclusion we used the most pessimistic scenario within the main body of the paper, which was high schools transmission $R_S = 1.5$. The maximum likelihood estimates for parameters using the high schools transmission scenario are given in main *Appendix 3—table 3*, and the maximum likelihood estimates for all scenarios summarised in *Appendix 3—figure 3*.

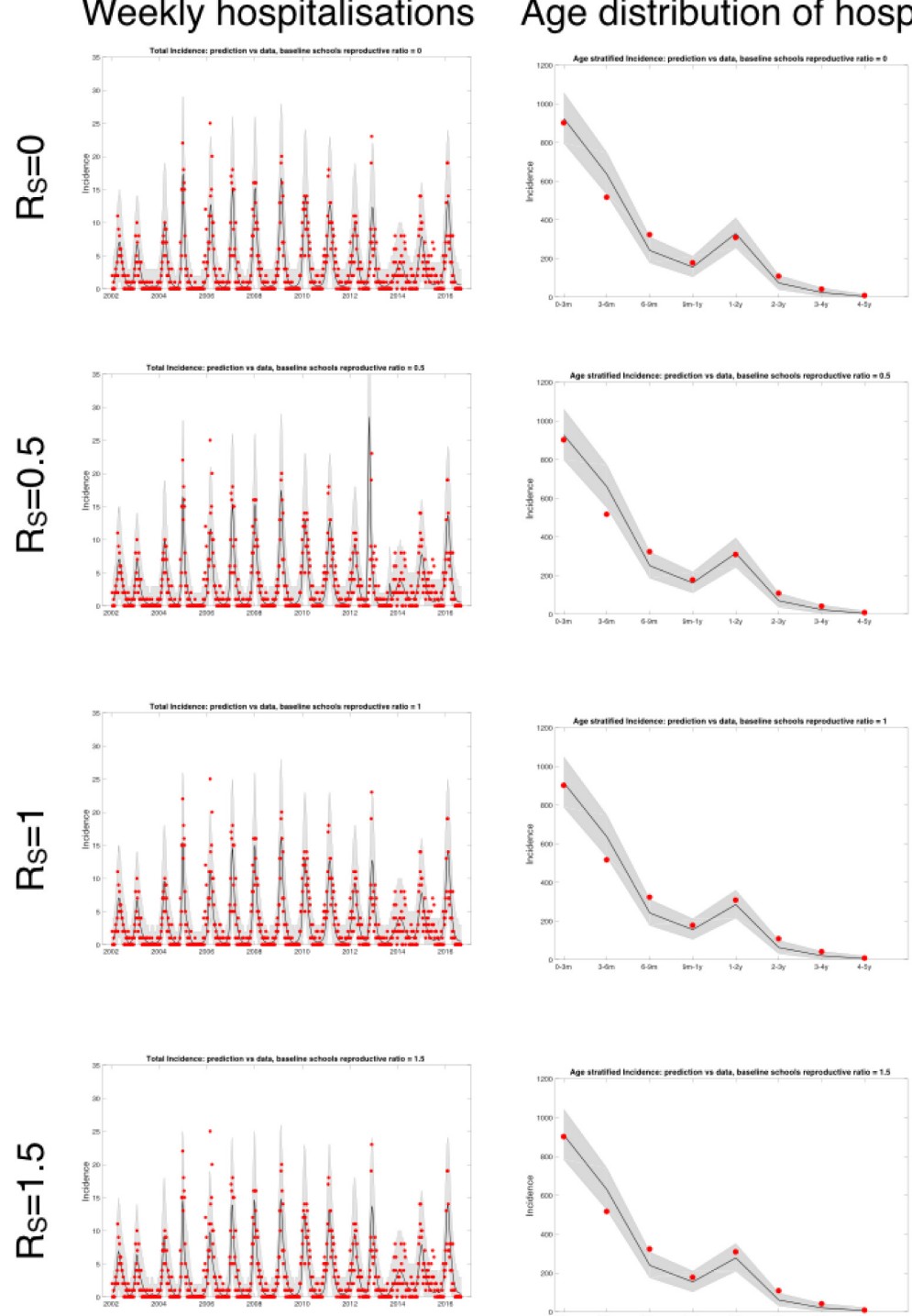

**Appendix 3—figure 2.** Plots of fitted weekly hospitalisations and the age distribution of hospitalisations for four scenarios (differing values of the schools based baseline $R_S$). In each case, parameter inference was performed and the maximum likelihood estimators used.

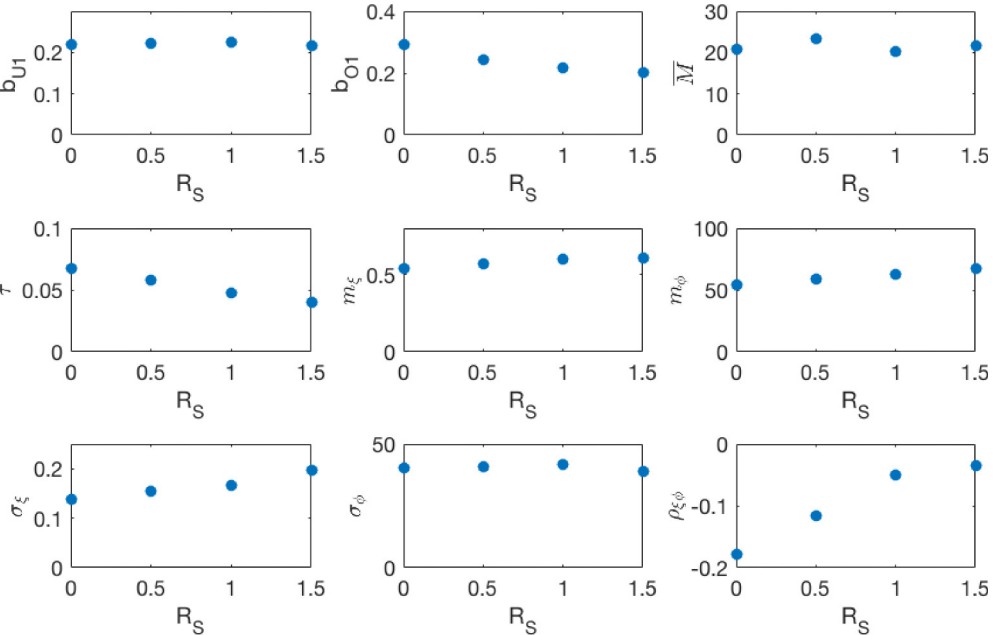

**Appendix 3—figure 3.** Maximum likelihood parameters for the different school transmission rate scenarios. $b_{U1}$, $b_{O1}$ are respectively the under-one and over-one mixing components of the community mixing rate matrix. $\tau$ is the rate at which a household member infectiously contacts *each* other household member. $\overline{M} = 1/\alpha$ is the mean period of maternal protection after birth. $m = (m_\xi \sim m_\phi)$ is the mean vector of the random seasonality, and $\sigma_\xi$, $\sigma_\phi$ and $\rho_{\xi\phi}$ are respectively the standard deviations of the seasonal amplitude, seasonal phase and the correlation between the two, derived from the estimated covariance matrix $\Sigma_{\xi\phi}$.

**Appendix 3—table 3.** Model parameters inferred from hospitalisation data.

| $b_{U1}$ | Community transmission rate for U1s | 0.22 [0.18,0.27] per day |
|---|---|---|
| $b_{O1}$ | Community transmission rate for O1s | 0.20 [0.18,0.21] per day |
| $\tau$ | Transmission rate to each other member of household | 0.040 [0.032, 0.048] per day |
| M | Mean duration of maternal protection at birth | 21.6 [17.2, 26.1] days |
| $m_\xi$ | Mean amplitude of log-seasonality | 0.61 [0.51, 0.72] |
| $m_\phi$ | Mean timing of log-seasonality peak (phase) | 67.7 [40.2, 77.7] days |
| $\sigma_\xi$ | Std. amplitude of log-seasonality | 0.20 [0.098,0.31] |
| $\sigma_\phi$ | Std. timing of log-seasonality peak (phase) | 38.7 [30.0, 48.5] days |
| $\rho_{\xi\phi}$ | Corr. coefficient between log-seasonal amplitude and phase | -0.035 [-0.12, 0.072] |

## Appendix 4

### Modelling vaccination in the household- and age-structured RSV transmission model

As described in the main paper we modelled the use of two different vaccines: a vaccine deployed to boost the period during which a newborn is protected from RSV by an unknown period $P$ with coverage $V_{cov}$ [MAB vaccine], and a vaccine deployed to O1 household members of the newborn which provokes a period of protection to RSV infection similar to the immunity period of a natural infection at household coverage $H_{cov}$ [IRP vaccine]. Already infected or recovered O1s were not affected by the IRP vaccine. We assumed that the MAB and IRP vaccines were deployed independently, which is useful for gauging potential effectiveness, but unrealistic. In reality, any reason a mother-to-be might miss being MAB vaccinated would also be a reason that the household O1s wouldn't get vaccinated.

The IRP vaccine altered the effective birth events by also provoking transitions to $R_2$ state at the point of birth,

Demographic turnover due to births with vaccination:

$$[s_1, i_1, r_1, s_2, i_2, r_2] \rightarrow [s_1 + 1, i_1, r_1, s_2 - 1, i_2, r_2] \text{ at rate}: (1 - H_{cov})\mu(n,t)s_2, \tag{64}$$

$$[s_1, i_1, r_1, s_2, i_2, r_2] \rightarrow [s_1 + 1, i_1, r_1, s_2, i_2 - 1, r_2] \text{ at rate}: (1 - H_{cov})\mu(n,t)i_2, \tag{65}$$

$$[s_1, i_1, r_1, s_2, i_2, r_2] \rightarrow [s_1 + 1, i_1, r_1, s_2, i_2, r_2 - 1] \text{ at rate}: (1 - H_{cov})\mu(n,t)r_2, \tag{66}$$

$$[s_1, i_1, r_1, s_2, i_2, r_2] \rightarrow [s_1 + 1, i_1, r_1, 0, i_2, s_2 + r_2 - 1] \text{ at rate}: H_{cov}\mu(n,t)(s_2 + r_2), \tag{67}$$

$$[s_1, i_1, r_1, s_2, i_2, r_2] \rightarrow [s_1 + 1, i_1, r_1, 0, i_2 - 1, s_2 + r_2] \text{ at rate}: H_{cov}\mu(n,t)i_2. \tag{68}$$

The MAB vaccine altered both the probability that an U1 is protected, and the age distribution of those who are infected. We denote the random period of time a newborn born to a MAB vaccinated mother is protected from RSV as $M_{vac} = M + P$, which has distribution function,

$$\mathbb{P}(M_{vac} \leq a) = \begin{cases} 0 & 0 \leq a \leq P \\ (1 - \exp(-(a-P)/\bar{M}))/(1 - \exp(-(T-P)/\bar{M})) & P \leq a \leq T \\ 1 & \text{otherwise} \end{cases} \tag{69}$$

The mean susceptibility of U1s after MAB vaccination has been applied to the population was,

$$\begin{aligned} \sigma_{U1,vac} &= \frac{1}{T}\int_0^T ((1 - V_{cov})\mathbb{P}(M \leq a) + V_{cov}\mathbb{P}(M_{vac} \leq a)) \sim da \\ &= 1 - \frac{\bar{M}}{T} + (1 - V_{cov})P\frac{e^{-T/\bar{M}}}{1 - e^{-T/\bar{M}}} + V_{cov}\frac{(T-P)e^{-(T-P)/\bar{M}}}{T(1 - e^{-(T-P)/\bar{M}})} - V_{cov}\frac{P}{T}. \end{aligned} \tag{70}$$

The conditional age category of an U1 who has definitely been infected, where $a = (a_0, a_1)$, after MAB vaccine has been deployed at coverage $V_{cov}$ was,

$$\begin{aligned} \mathbb{P}(A \in a | \tilde{M} < A, A \leq 1 \text{ year}) &= 1(a \leq 1 \text{ year}) \\ &\quad \frac{((1-V_{cov})\mathbb{P}(M < A | A \in a) + V_{cov}\mathbb{P}(M_{vac} < A | A \in a))\mathbb{P}(A \in a | a \leq 1 \text{ year})}{\mathbb{P}(M < A | a \leq 1 \text{ year})} \\ &= \frac{1(a \leq 1 \text{ year})}{T\sigma_{U1,vac}}\left((1 - V_{cov})\frac{a_1 - a_0 + \bar{M}(e^{-a_1/\bar{M}} - e^{-a_0/\bar{M}})}{1 - e^{-T/\bar{M}}} + V_{cov}f(a,P)\right). \end{aligned} \tag{71}$$

where $\tilde{M}$ is the random maternal protection duration of a newborn before we observe whether the newborn's mother had been MAB vaccinated. The function $f(a,P)$ completes *Equation (71)* by giving the age distribution of U1s who had boosted maternal protection to RSV but was nonetheless infected,

$$f(a,P) = \begin{cases} 0 & a_0 \le P \text{ and } a_1 \le P \\ \frac{a_1 - P + \overline{M}(e^{-(a_1-P)/\overline{M}} - 1)}{1 - e^{-(T-S)/\overline{M}}} & a_0 \le P \text{ and } a_1 > P \\ \frac{a_1 - a_0 + \overline{M}(e^{-(a_1-P)/\overline{M}} - e^{-(a_0-P)/\overline{M}})}{1 - e^{-(T-S)/\overline{M}}} & a_0 > P \text{ and } a_1 > P \end{cases} \tag{72}$$

Note that because $\sigma_{U1,vac}$ depended on $V_{cov}$ the age distribution of infected U1s depended on $V_{cov}$ in a nonlinear fashion.

We considered a range of values for $P$ and $H_{cov}$ for each of the schools transmission scenarios; using the maximum likelihood estimators for the inferred parameters for each scenario. In each scenario, at $V_{cov} = 1$ the median reduction in hospitalisations was similar, although for the high school transmission scenario vaccination was slightly less effective (*Appendix 4—figure 1* and *Appendix 4—figure 2* colorblind-friendly version ). Therefore, we used this scenario in the main paper as a pessimistic/robust example. As mentioned in main text we simulated 10 years into the future over 500 independent realisations of the random seasonality. Presented are medians of % reduction in hospitalisations at KCH compared to no intervention.

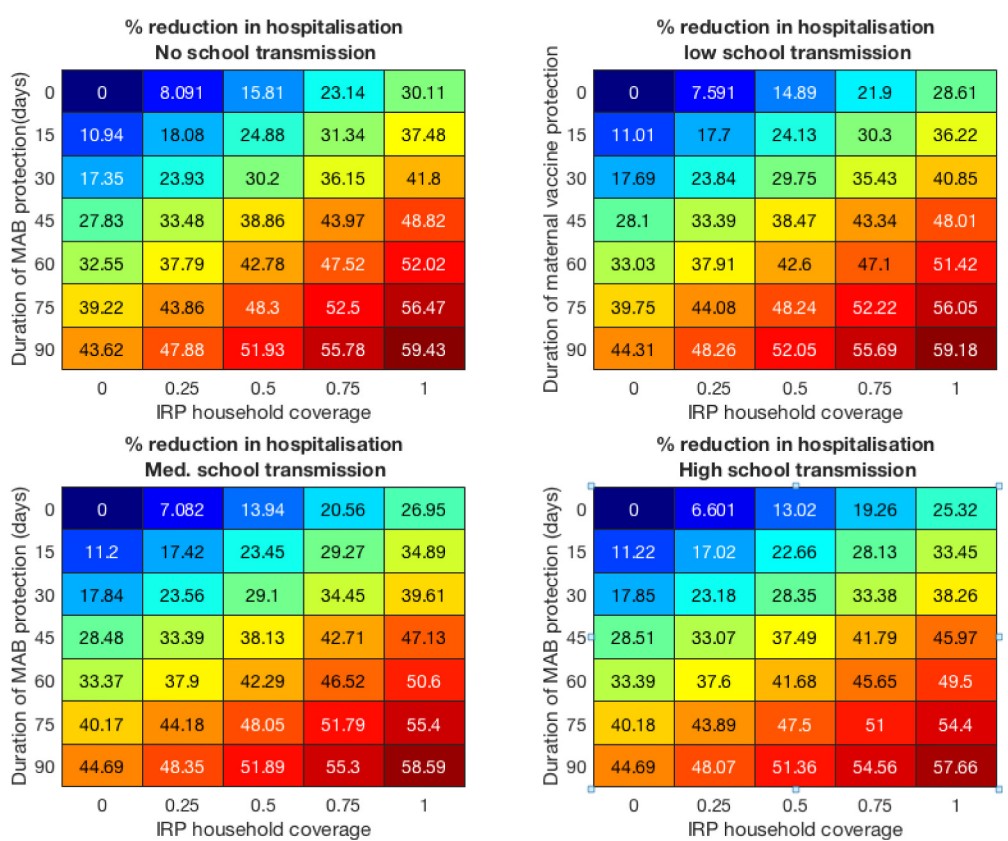

**Appendix 4—figure 1.** Vaccine effectiveness for the four school mixing scenarios at 100% MAB coverage.

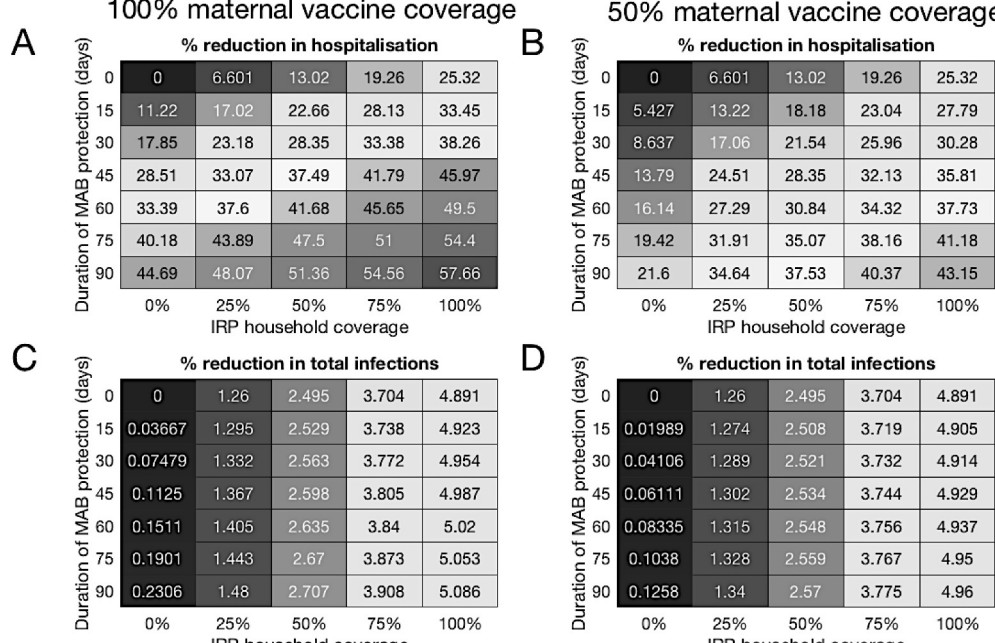

**Appendix 4—figure 2.** Colorblind-friendly version of *Figure 4* from main text. Forecast effectiveness of RSV vaccination for different mixed strategies over a 10 year period for 100% maternal vaccine effective coverage (**A and C**) and 50% maternal vaccine effective coverage (**B and D**). (**A and B**) Percentage reduction in hospitalisations at KCH. (**C and D**) Percentage reduction in total RSV infections in the population.

