## [Decision Letter]

**Acceptance summary:**

Brand and colleagues present an agent-based model of respiratory syncytial virus transmission and vaccination and use it to explore potential vaccination schedules in pregnant women. They find up to 50% reductions in infant RSV infections with fairly high coverage of the prenatal vaccine. This work is important for informing future vaccination policy when RSV vaccines become available.

**Decision letter after peer review:**

Thank you for submitting your article "Reducing RSV hospitalisation in a lower-income country by vaccinating mothers-to-be and their households" for consideration by *eLife*. Your article has been reviewed by Neil Ferguson as the Senior Editor, a Reviewing Editor, and two reviewers. The following individuals involved in review of your submission have agreed to reveal their identity: Katherine Atkins (Reviewer #2).

The reviewers have discussed the reviews with one another and the Reviewing Editor has drafted this decision to help you prepare a revised submission.

Summary:

In the current study Brand et al. use a mathematical model to estimate reductions in RSV hospitalizations in children following a theoretical maternal and >1 y/o vaccines. The study builds on much previous work conducted by members of the same group and adds an important dimension to the question surrounding RSV epidemiology and the potential for vaccination.

The authors find modest reductions in overall hospitalizations but a sizable reduction in hospitalizations in children <1 y/o – the most vulnerable group. It is encouraging to see the modelling being conducted with an LMIC setting too, as most previous RSV modelling efforts have been HIC-based. The modeling methods are sound (and elegant) but the manuscript would benefit from some clarification of both methods and results.

Essential revisions:

1) I think it would be helpful if more motivation were to be provided on the reasoning behind the household model choice. If I understand correctly, there is a serious computational downside of solving these types of models, at the expense of some epidemiological realism (with respect to neglecting exposure-dependent parameters, for instance). I'm not advocating the authors conduct a comparison, but I'm interested to know whether the choice of household model can ultimately reflect the impact of household-based strategies more accurately than a more epidemiologically-realistic model can using approximations for the mother-child contact (for this, see again, Atkins et al., 2016).

2) The paper needs to be grounded in relation to vaccines currently being developed. There is no discussion as of now of potential future vaccines and there should be to motivate the work.

3) From what I understand, the contact within the household is assumed to be density dependent, whereas the contact outside of the household is assumed to be frequency dependent – is this right? Could you mention this?

4) Hasn't there been mixing matrices conducted in Kilifi that could be used?

5) It is a big assumption that the demographics are fixed in the 10 years of prediction. The birthrate has been declining in Kenya over the past 15 years (see https://www.indexmundi.com/g/g.aspx?c=ke&v=25). Thus, the reported reductions may be over-estimated.

6. I'd like to have a comprehensive parameter table that describes all parameters used (vaccine duration, uptake etc. – including fitted parameters). This would help in understand the base case scenario – which was difficult to find as well as the reliability of the model and deviation from previous work.

7) Figure 2 suggests to me that the model underestimates the number of hospitalisations. It would be useful to see the absolute numbers stratified by age (rather than just by% ).

8) Figure 5A: It is extremely surprising to me that the post-vaccine dynamics immediately equilibrate – do the age distribution infection also equilibrate immediately? Presumably, but this is also surprising.

9) Figure 6: I can't decipher what these combined strategies are – we need more information in the caption accompanied with a separate table that spells out which vaccine strategies are being considered. It is very difficult to interpret the differences between the strategies otherwise.

10) Does Figure 6 really report avoided hospitalizations? If so, why do avoided hospitalizations go down with increasing coverage? Also, I would suggest reversing the order of the legend to match the order of the lines.

11) Introduction: I'm not sure I agree with this assessment. Admittedly in the context of pertussis, but nevertheless, our modelling study suggested the benefit of cocooning in the presence of direct protection of the infant was extremely marginal (Atkins et al., 2016). That is, in terms of impact, there was no point in cocooning when substantial direct protection had been achieved.

12) Results section: Where are the derivations for the equations in box 1? They are not immediately obvious.

13) Results section: I'm a little sceptical at both the values of R0 and the conclusions drawn, namely that community transmission has an R0 < 1 (on average an infection initiated at random) produces <1 other case in the community). This is because when R0 is calculated from a model, the structure of the model, as the authors note, can substantially impact the value of R0 calculated. While the authors note the difference between age and household related structure, the number of exposure classes will also make a difference I think. Perhaps the authors can comment on this.

14) Materials and methods section: Originally you said you split up <1 and >1 years, but there is discussion of finer age groups here, this needs to be explained more clearly as I'm confused with the age stratification, its parameterisation and how it was implemented in the model.

15) Subsection “Conditional age of individuals”: When you say 'we calculated empirical distributions', it's not clear how you constructed these distributions, or how they were 'empirical' – more information please. Presumably there are parameterised from the KDHSS survey, but it's not clear. More link between the data and the distributions needed I think.

16) Subsection “Hospitalisation rates”: I think I understand what the authors are trying to do here – there needs a little bit of a fudge to account for the exposure-dependent nature of RSV infection that is missing from the model. However, there are two forces at play here, which I think need to be captured independently. First, is the age-specific nature of infection – evidence points to severe infection / hospitalisation being necessarily age dependent, when the lung pathways are not fully developed and infected infants are more at risk of bronchiolitis than their older counterparts. (arguably very young infants are also more likely to be picked up in surveillance through increased testing and reporting). There is then the exposure-dependent nature of infection, that is the higher chance of asymptomatic infection with increasing exposures. Thus, with passive protection from either extended life monoclonals or by maternal vaccination, the idea is to push infants out of their most risky period (age-dependent severe infection), with the trade-off that no vaccine- or natural-immunity is elicited and they still have the same risk of symptomatic infection as younger individuals. If I understand correctly, the model captures the latter mechanism, but not the first. More clarity is needed on distinguishing these phenomenons I think.

---

## [Author Response]

Essential revisions:1) I think it would be helpful if more motivation were to be provided on the reasoning behind the household model choice. If I understand correctly, there is a serious computational downside of solving these types of models, at the expense of some epidemiological realism (with respect to neglecting exposure-dependent parameters, for instance). I'm not advocating the authors conduct a comparison, but I'm interested to know whether the choice of household model can ultimately reflect the impact of household-based strategies more accurately than a more epidemiologically-realistic model can using approximations for the mother-child contact (for this, see again, Atkins et al., 2016).

We have now expanded on this motivation in the main text (Introduction).

The additional computational complexity of simulating the age-and-household model used in this paper, compared to an age-structured model using an age-specific contact matrix, was a significant challenge. The advantage of explicit inclusion of household structure in the model is that the social contacts within the household are persistent over multiple RSV seasons, whereas age-structured models implicitly assume random mixing within the confines of the mixing matrix (that is all people of a given age group are equally likely to be contacted by any individual at any instant and therefore the chance of repeated contact become zero as the population size becomes large). In short, the age-and-household model used in this paper is a special type of contact network model whereas the standard age-structured model is effectively a random mixing model. Therefore, the usefulness of the household model depends on whether the network clustering of social contacts is important for simulating RSV transmission, and in particular for estimating the risk of transmission to under one year olds.

In the specific case of modelling highly seasonal RSV transmission, we would argue that a network-like transmission structure is important for capturing the relevant epidemiology. Most people have caught RSV by the age of two and will have multiple repeated episodes during their lifetime. The time between recovery from an episode and reversion back to at least partial susceptibility is estimated to be ~6 months. In Kilifi county, there are sharp annual peaks of RSV hospitalisation at each seasonal RSV epidemic, and so one should expect the population to consist of large numbers of entirely susceptible and partially susceptible individuals due to the inter-epidemic period being longer than the typical time over which loss of immunity to RSV occurs. These general considerations suggest that (i) RSV seasonal epidemics will be akin to repeated invasions of a nearly susceptible population, i.e. closer to the “epidemic” scenario than an “endemic” scenario, and (ii) RSV transmission is much closer to the “SIS” than the “SIR” paradigm. Network effects are most important during an epidemic invasive growth phase (Miller 2009) and are typically more important for SIS-type dynamics with persistent contacts (Sun, Baronchelli and Perra, 2015). Both these features appear to be important for seasonal RSV transmission in Kilifi and therefore provide strong motivation for the network-type epidemic model we have used.

Another motivation for using detailed household structure in an epidemic model for Kilifi county was the availability of detailed demographic data covering the period over which we also have hospitalisation data. Our model was parametrised using the joint distribution of household type and age of household inhabitants; this allowed us to include age-related effects (such as the age-dependent probability of hospitalisation conditional on being infected) by reference to the age distribution for the type of household in which infection occurs. This would not be an effective modelling approach for infectious diseases with long periods of immunity (i.e. decades) where the age-distribution of susceptibility is a highly important factor. However, for the specific case of highly seasonal RSV we argue that this sufficiently captures the age distribution of susceptibles whilst allowing us to account for network-like social structure.

We consider the mother-child approximation used in Atkins et al. very appropriate for modelling pertussis in a high-income country with household sizes typically smaller than Kilifi, but believe it would be insufficient for modelling RSV in our setting. In a household containing a newborn one might also expect one or two parents, and possibly older siblings of the newborn. The length of time over which immunity to pertussis wanes after either a natural infection or routine childhood vaccination is believed to be in the order of decades. Therefore, the older siblings cohabiting with the newborn will have a reasonable probability of being immune (after either vaccination or natural infection) and not contribute to transmission within the household. Therefore, ignoring social clustering in the household is reasonable since a number of members of the household cluster cannot contract or transmit. This argument is supported by simulation studies comparing SIR type transmission both with and without explicit household structure (Glass et al., 2013); that is that the differences in long-term incidence between the two model types were marginal for most age groups when parametrised using the same demographic data. However, as we have argued above, specifically for modelling seasonal RSV this approximation is inadequate because most households will have a number of at least partially susceptible members, and therefore, social clustering is a relevant factor in epidemic prediction.

2) The paper needs to be grounded in relation to vaccines currently being developed. There is no discussion as of now of potential future vaccines and there should be to motivate the work.

The opening paragraph of the Introduction refers the reader to WHO goals for potential RSV vaccines and a summary document of current vaccines and their development stage. We have also presented our results in light of the partially successful ResVax trail.

These provide justification for investigating the option of boosting infant antibody through maternal immunization (or long-lasting monoclonal at birth). Motivation for the option of vaccinating household co-habitants of mothers-to-be, arises in part from the same sources, which identify the paediatric population as the second key target group for vaccine development. We are further influenced by epidemiological studies suggesting that elder family members such as school age children or parents are possible sources introducing RSV into households leading to infant infection (Anderson et al., 2013; Graham, 2014). Of the current vaccines under development, temporary immunity to RSV in these older (seropositive) household individuals would most likely be achieved by a sub-unit vaccine (Anderson et al., 2013). We have updated the text and included the additional reference to Graham, 2014.

3) From what I understand, the contact within the household is assumed to be density dependent, whereas the contact outside of the household is assumed to be frequency dependent – is this right? Could you mention this?

This is correct, we assume density dependent transmission within the household. We have corrected the paper to mention this explicitly (subsection “Model Dynamics, forces of infection and susceptibility to RSV”).

4) Hasn't there been mixing matrices conducted in Kilifi that could be used?

The main paper that has used mixing matrices for Kilifi was Kinyanjui et al., 2015. Two different mixing matrices were considered each derived from a separate data source: (i) a contact diary study, and (ii) co-occupancy as recorded in the Kilifi health and demographic surveillance system (KHDSS) over a snapshot of Sept 2010 – Jan 2011 (Scott et al. 2012). We couldn’t construct a reliable estimate of the full social contact graph from 568 diary responses, therefore we used the KHDSS co-occupancy data to construct a social contact graph of household cohabitants, and assumed that other social contacts, for example at schools, occurred as age-structured random mixing.

Our approach to using the same KHDSS data differed from Kinyanjui et al., 2015 in two ways: first, they used household co-occupancy to create an age-structured mixing matrix, effectively deconstructing the social contact network, whereas we maintain the social contact network for households, and second, they used a single snapshot of the KHDSS sample population, whereas we used multiple snapshots and dynamically evolved the household structure between snapshots.

We have made our approach clearer in a new subsection “Joint distributions of age and household occupancy”.

5) It is a big assumption that the demographics are fixed in the 10 years of prediction. The birthrate has been declining in Kenya over the past 15 years (see https://www.indexmundi.com/g/g.aspx?c=ke&v=25). Thus, the reported reductions may be over-estimated.

We have been sensitive to incorporating known historic changes in demography into our inference method, both in terms of numbers of individuals available to be infected and their joint age-and-household type distribution. This had the benefit that we could disentangle seasonal variation in hospitalisation from simply having a larger number of individuals at risk of hospitalisation.

However, for forecasting, we felt that it was reasonable to consider an idealised demographically static population with results presented in terms of percentage reduction of hospitalisation. Our reasoning was that although the crude per-capita birth rate has declined in Kenya as the population size has increased, the size of the critically at-risk under one-year-old population in Kilifi has stabilised at roughly 8500. Also, in forecasting demographic change in our model we would need to consider combinations of factors. For example, if there was a declining number of critically at-risk newborns but also an increase in the typical number of individuals per household due to increasing absolute population size we might expect hospitalisations to increase due to increasing transmission risk per newborn, despite the number of newborns at-risk decreasing. This would be an interesting avenue of investigation for future work, but not appropriate for this paper. We have added an explicit discussion on this into the Discussion section.

6. I'd like to have a comprehensive parameter table that describes all parameters used (vaccine duration, uptake etc. – including fitted parameters). This would help in understand the base case scenario – which was difficult to find as well as the reliability of the model and deviation from previous work.

We have now added a table of vaccination scenarios (Table 1) and moved both the literature derived parameter table (Table 2) and the inferred parameters table (Table 3) into the main text from the supporting information.

7) Figure 2 suggests to me that the model underestimates the number of hospitalisations. It would be useful to see the absolute numbers stratified by age (rather than just by% ).

The prediction of the model, after parameter inference, is that the total number of hospitalisations would be Poisson (2271) distributed whereas the true number of hospitalisations was 2382 (4.7% relative error compared to mean prediction).

However, most of the error is concentrated in the outlier RSV epidemic during 2005-2006. This was the only year which had three pronounced peaks in hospitalisations at least a month apart: two smaller peaks on 11th Dec 2005 and 24th Mar 2006 around a larger peak on 24th Feb 2006. The model was unable to capture this unusual temporal pattern in hospitalisation.

With the 2005-2006 RSV season excluded (all datapoints from 4th Nov 2005 to 9th June 2006) the model predicts an average of 2147 hospitalisations over the rest of the period compared to 2174 actual hospitalisations (1.2% relative error). This is now discussed in Results section.

Figure 2 has been altered to present absolute numbers.

8) Figure 5A: It is extremely surprising to me that the post-vaccine dynamics immediately equilibrate – do the age distribution infection also equilibrate immediately? Presumably, but this is also surprising.

We were initially surprised by the rapid shift in dynamics as well (it is also true for the age distribution). However, in the context of seasonal RSV transmission and the specific vaccination strategies considered in the paper, we find the rapid shift understandable. First, even with both vaccination types used at the maximum effectiveness and coverage considered in the paper the reduction in total RSV infections is slight (<4% reduction compared to no vaccination in Figure 5A). Therefore, despite a significant reduction in hospitalisation, the overall epidemic should not be thought of as having been pushed far from its previous dynamics. Second, as mentioned above, seasonal RSV is closer to the “SIS” paradigm of disease rather than the “SIR” paradigm. Therefore, a perturbation away from equilibrium does not necessarily cause additional oscillatory dynamics (typical of perturbed SIR models) unless we get close to eradication. We noted that in Kinyanjui et al., 2015for their age-structured model based on household co-occupancy, which can be thought of as the age-structured “version” of our model, there is also rapid transition from high rates of hospitalisation pre-vaccination to lower levels post-vaccination. We mention the rapid change as notable on Results section, as well as discussing the more SIS-type dynamics in the Introduction.

9) Figure 6: I can't decipher what these combined strategies are – we need more information in the caption accompanied with a separate table that spells out which vaccine strategies are being considered. It is very difficult to interpret the differences between the strategies otherwise.10) Does Figure 6 really report avoided hospitalizations? If so, why do avoided hospitalizations go down with increasing coverage? Also, I would suggest reversing the order of the legend to match the order of the lines.

Reply to (9) and (10): The purpose of Figure 6 was to demonstrate the efficiency of vaccination at different levels of coverage; that is either reduced numbers of hospitalisations or infections per vaccine. However, we agree that its current format leaves the reader unsure about the message of the plot. We have reworked this plot for additional clarity. We have redone Figure 6 to remove the dashed lines and give the efficiency results in stacked bar plot format, as well as redoing the caption to reference Table 1. We have also substantially rewritten the paragraph explaining these findings (Results section).

11) Introduction: I'm not sure I agree with this assessment. Admittedly in the context of pertussis, but nevertheless, our modelling study suggested the benefit of cocooning in the presence of direct protection of the infant was extremely marginal (Atkins et al., 2016). That is, in terms of impact, there was no point in cocooning when substantial direct protection had been achieved.

We agree that if substantial direct protection for infants from RSV could be achieved then additional measures would not be necessary, e.g. a vaccine that gave substantial protection from birth for first two years of life would dramatically reduce RSV hospitalisations irrespective of other factors.

However, it is not yet clear whether such a highly effective vaccine will become available; the results of the Novavax ResVax maternal vaccine were mixed indicating only partial success in direct protection of infants. One outcome of our modelling study is to demonstrate that a mixed strategy of partially effective vaccines could be complementary. We have adjusted our Discussion section to make this clearer. This is mentioned in the Introduction.

We would caution against drawing too much analogy with pertussis in the USA. Essentially, ‘cocooning’ has already occurred in that context since high coverage and effectiveness of childhood pertussis vaccination already exists. As argued above, in a typical household containing a newborn and one or two parents in the USA the most likely other household co-habitants would be older siblings of the newborn, and due to (DTaP) vaccination coverage these older siblings have a high probability of being immune to transmission. In this context, one might expect 6 months of effective direct protection given to a newborn by a booster (Tdap) vaccine for the mother, as per Atkins et al., 2016 to be sufficient.

We have re-worded this section to make the comparison (and differences) with pertussis clear (Introduction).

12) Results section: Where are the derivations for the equations in box 1 (Results section)? They are not immediately obvious.

This was an oversight; we have added a complete derivation to the supporting information. Also, there was a typo in the relative reduction result as presented in Box 1 which has been corrected.

13) Results section: I'm a little sceptical at both the values of R0 and the conclusions drawn, namely that community transmission has an R0 < 1 (on average an infection initiated at random) produces <1 other case in the community). This is because when R0 is calculated from a model, the structure of the model, as the authors note, can substantially impact the value of R0 calculated. While the authors note the difference between age and household related structure, the number of exposure classes will also make a difference I think. Perhaps the authors can comment on this.

This section was mis-judged. We calculated the R0 value from a next-generation matrix based on the inferred transmission rates. However, the transmission rates were inferred jointly with the seasonality parameters so analysing them independently was a poor decision. We have now removed this paragraph it did not add substantially to the main message of the paper.

The relationship between exposure classes and R0 for RSV is interesting because it is currently believed that individuals are typically less susceptible, less infectious and have a shorter infectious duration on their second and subsequent exposure to RSV infection. However, R0 is calculated with reference to an idealised completely naïve population; that is that everyone in the idealised population has never been exposed. Therefore, threshold R0 only depends on the susceptibility and infectiousness of the first exposure class.

In principle, this could mean that the first exposure class (effectively newborns and young children) is critical to maintaining RSV in the population, since it is possible that RSV could not be sustainable amongst only people in multiple exposure classes. However, it seems unlikely that a virus with seemingly high prevalence in every population is being sustained by such a small group. Whilst this is an interesting aspect, we haven’t added to the main manuscript to reflect this discussion because we feel it would dilute from focus of this paper: estimating transmission rates and therefore the potential impact of a specific set of vaccination strategies.

14) Materials and methods section: Originally you said you split up <1 and >1 years, but there is discussion of finer age groups here, this needs to be explained more clearly as I'm confused with the age stratification, its parameterisation and how it was implemented in the model.

This was a poorly described aspect of the model. From a dynamical point of view we use a simple <1 and >1 year old age classification, however within those groups we get richer detail by considering what age a <1/>1 year old is likely to be given the household they live within.

In brief, age stratification is implicit within our modelling framework. We know the age-structure within all households in the KHDSS area with children under and over one – therefore we can relate any infection in a household to potential ages. This is used to both derive age-dependent infection probabilities and age-structured mixing between households.

In this modelling study, we invert the usual approach to averaging over epidemiologically important quantities for “SIR” type infection models. For infectious pathogens that provoke long-lasting or permanent immunity the age-distribution of susceptibility is a critical dynamic component, and therefore, it makes sense to model age structure dynamically and average over social structure (i.e. household cohabitation). For seasonal RSV, we expect a large fraction of the population will be at least partially susceptible at the beginning of each epidemic. In this setting, we believe that age structure is less important than social structure to the dynamics of transmission. Consequently, we model household structure explicitly and consider the age of individuals, conditional on their household and whether they are known to be under-one or over-one. This allowed us to estimate age-dependent quantities such as the risk of hospitalisation without having to include them as explicit dynamic model variables.

We feel that this was an innovative aspect to our modelling approach, which we have not explained clearly. We have added an extra subsection on this aspect in the model description (subsection “Joint distributions of age and household occupancy”).

15) Subsection “Conditional age of individuals”: When you say 'we calculated empirical distributions', it's not clear how you constructed these distributions, or how they were 'empirical' – more information please. Presumably there are parameterised from the KDHSS survey, but it's not clear. More link between the data and the distributions needed I think.

As the reviewer guessed this was derived by linking individuals presents in the KHDSS surveys by their house ID number. We have substantially increased our description of this in the paper subsection “Joint distributions of age and household occupancy” and Appendix.

16) Subsection “Hospitalisation rates”: I think I understand what the authors are trying to do […] If I understand correctly, the model captures the latter mechanism, but not the first. More clarity is needed on distinguishing these phenomenons I think.

It is correct that the main protective effect of the maternal/extended life monoclonal vaccine is to shift the age of first infection away from the critical early months of life. However, we argue that our model is better at capturing age-specific effects of infection, and therefore possible hospitalisation compared to exposure-dependent effects. Because a significant percentage of children have caught RSV by the age of one, and a large majority by the age of two, it is hard to disentangle exposure vs age as risk factors for severe disease due to RSV. However, there is some evidence that age is the more critical factor (Ohuma et al., 2012).

As mentioned above, we assume that individuals, within the crude under-one/over-one dynamic groups, have an age distribution which depends on their household size and whether the household contains an under-one year old or not. This allowed us to capture social structure aspects such as households with an under-one also typically having school age children (if the household size is > 3) which potentially alters the risk of RSV introduction.

The exposure-dependent nature of infection is covered by the approximation that all over-ones have been infected at least once by RSV. A large majority of all people over the age of two have contracted RSV, so only the 1-2 year olds are likely to be significantly misrepresented by this assumption. If a truly effective RSV vaccine became available, this approximation would become increasingly problematic as the average age of first infection increased, however for the scenarios investigated in this paper we consider this approximation reasonable.